# Spying on parahydrogen-induced polarization transfer using a half-tesla benchtop MRI and hyperpolarized imaging enabled by automation

Frowin Ellermann [1,4], Aidan Sirbu [2,4], Arne Brahms [3], Charbel Assaf[1], Rainer Herges [3], Jan-Bernd Hövener [1] & Andrey N. Pravdivtsev [1] ✉

Nuclear spin hyperpolarization is a quantum effect that enhances the nuclear magnetic resonance signal by several orders of magnitude and has enabled real-time metabolic imaging in humans. However, the translation of hyperpolarization technology into routine use in laboratories and medical centers is hampered by the lack of portable, cost-effective polarizers that are not commercially available. Here, we present a portable, automated polarizer based on parahydrogen-induced hyperpolarization (PHIP) at an intermediate magnetic field of 0.5 T (achieved by permanent magnets). With a footprint of 1 m², we demonstrate semi-continuous, fully automated $^1$H hyperpolarization of ethyl acetate-*d6* and ethyl pyruvate-*d6* to P = 14.4% and 16.2%, respectively, and a $^{13}$C polarization of 1-$^{13}$C-ethyl pyruvate-*d6* of P = 7%. The duty cycle for preparing a dose is no more than 1 min. To reveal the full potential of $^1$H hyperpolarization in an inhomogeneous magnetic field, we convert the anti-phase PHIP signals into in-phase peaks, thereby increasing the SNR by a factor of 5. Using a spin-echo approach allowed us to observe the evolution of spin order distribution in real time while conserving the expensive reagents for reaction monitoring, imaging and potential in vivo usage. This compact polarizer will allow us to pursue the translation of hyperpolarized MRI towards in vivo applications further.

Nuclear magnetic resonance (NMR) is a powerful technique that continuously finds impressive applications in a number of scientific fields. Its notoriously low sensitivity has spurred the development of NMR and MRI instruments with high magnetic fields (very cost-intensive), sophisticated electronics, elaborate imaging technologies, and hyperpolarized media. Using dissolution dynamic nuclear hyperpolarization (dDNP)[1], pyruvate, glucose, and many other molecules have been hyperpolarized to about 50% at the $^{13}$C nucleus. In contrast, at thermal equilibrium and at a magnetic field of 3 T, the polarization of $^{13}$C nuclear spin is merely 0.00026%. The gain in polarization results in a proportional signal gain and has enabled real-time in vivo observations of metabolism[2–4] and novel diagnostics, e.g., for cancer and therapy monitoring[5]. Pyruvate is particularly interesting for clinical applications due to its fast metabolic reduction into lactate in certain tumors[6].

[1]Section Biomedical Imaging, Molecular Imaging North Competence Center (MOIN CC), Department of Radiology and Neuroradiology, University Medical Center Kiel, Kiel University, Am Botanischen Garten 14, 24118 Kiel, Germany. [2]Western University, 1151 Richmond St, London, ON N6A 3K7, Canada. [3]Otto Diels Institute for Organic Chemistry, Kiel University, Otto- Hahn Platz 4, 24118 Kiel, Germany. [4]These authors contributed equally: Frowin Ellermann, Aidan Sirbu. ✉e-mail: andrey.pravdivtsev@rad.uni-kiel.de

Parahydrogen-induced polarization (PHIP)[7,8] is an alternative method to dDNP. The use of parahydrogen (pH$_2$) as a source of nuclear spin order to enhance the NMR signals has been instrumental in studying metalorganic complexes[9], heterogeneous catalysis[10,11], photo-switching systems[12,13], a number of hydrogen transfer reactions[14–16], and even for in vivo angiography[17]. With the development of PHIP sidearm hydrogenation (PHIP-SAH) and the respective pyruvate precursors[18–20], the translation of PHIP into biomedical applications has significantly advanced[21]. However, this is still a rather experimental technology that requires well-mastered manual manipulations.

Automated auxiliary equipment for NMR and MRI experiments has always led to significantly improved reproducibility and a simpli-fied workflow. Some examples are the automation of magnetic field cycling[22–24] (MFC), laser irradiation[25], supplies of gases[26,27], and circu-lation of chemicals[28].

While pH$_2$ is perfectly ordered, it has a total nuclear spin of 0 and is NMR silent. To make use of the pH$_2$ spin order for the hyperpolar-ization of a molecule, chemical addition[8] or chemical exchange with the target is needed[29]. This hydrogenation reaction (or chemical exchange) usually requires well-controlled conditions, such as pres-sure, temperature, and timing, possibly within an NMR unit (depend-ing on the chosen approach)[30].

Setups dedicated to carrying out the entire polarization process are referred to as polarizers. Often, the polarization process in such a device, e.g., dDNP, involves several steps, for example, sample cooling, microwave irradiation, fast dissolution, and ejection. The commercial availability of such polarizers was essential for the widespread of dDNP. Unfortunately, a commercial PHIP polarizer is not yet available.

In pursuit of the optimal polarizer, several approaches for PHIP were described, from shake and run experiments with an NMR tube[31], to complex, stand-alone[17,32] or in situ polarizers for MRI[33] or NMR[22,26,34–36].

The first PHIP polarizers for MRI contrast agents used strong spin-spin interactions between $^1$H and $^{13}$C during MFC at fields below 0.1 mT[37,38] to transfer spin order from pH$_2$ to $^{13}$C. Soon after, spin order transfer (SOT) pulse sequences and dedicated low-field polarizers were introduced[39], doubling the experimental $^{13}$C polarization to ~50% for partially deuterated 2-hydroxyethylpropionate, whereas only 21-25% was reached with MFC[40]. Hyperpolarized 2-hydroxyethylpropionate is not an endogenous, metabolically active molecule like pyruvate and was used for MRI angiography instead[37]. There are some more low-field systems with RF field induced SOT[41–44], which have recently been comprehensively reviewed[45].

Strong indirect interactions between different nuclei ($^1$H, $^2$H, $^{13}$C, etc.) limit the efficiency of SOT[46,47]. As a result, often, 100% polarization can not be achieved (even theoretically) with MFC[40,48,49]. Pulse sequences and irradiation-based SOTs are more flexible in this regard[50–52], offering higher polarizations but higher hardware demands, too.

So far, some of the highest PHIP-SAH polarizations (especially for selectively deuterated molecules) have been demonstrated using RF-induced SOTs inside superconducting high-field magnets[21,53,54]. Simu-lations also predict that the highest polarization can be achieved between 1 mT to 1 T (depending on the spin system). Fortunately, such fields can be realized easily, e.g., with portable permanent magnets or electromagnets with small footprints, low cost, and low maintenance (e.g., not requiring cryogens)[55,56]. The low-field magnets and reactor, which are scalable in size, can be adjusted to fit the desired volume of hyperpolarized media. In rodent studies, a typical injection dose is 150 µL, while in human studies, it can be as high as 30 mL. Achieving polarization of volumes used in human studies is challenging and has not yet been demonstrated with superconducting high-field polarizers.

One of the latest additions to the family of PHIP polarizers are reactors operated within high-field MRI systems, known as SAMBADENA[57]. While the hardware requirements for this approach are low, challenges include limited space for the reactor (and thus limited tracer volume), accessibility, safety, and availability of MRI-compatible materials.

Here, we present a PHIP polarizer with an automated sample shuttling and a pH$_2$ hydrogenation pressure of up to 30 bar. The sys-tem uses a commercially available, portable, half-tesla benchtop teaching MRI scanner based on permanent magnets (detailed in methods). This mobile MRI allows, by design, for a larger sample volume than NMRs, and features the necessary electronics for SOT, signal acquisition, automation of the experimental hyperpolarization procedure, and self-calibration. For example, the automatic $B_0$ field calibration was convenient when the system was moved to another lab site; the calibration took only about 2 min. In addition, the system's mobility makes it possible to place it wherever necessary (e.g., in close proximity to a preclinical MRI).

The device neither requires cryogens nor a particular power connection as it operates with a domestic power socket, minimizing requirements for maintenance and installation that lowers the barrier for the community to start using it. We found it convenient to polarize 300 µL solutions, although larger volumes are possible. In contrast, high-resolution, narrow-bore NMR systems allow only 100 – 200 µL to be polarized due to restrictions in the available volume, $B_0$ and $B_1$ homogeneity[21,47].

Although the low-field MRI system does not have a high spectral resolution (Fig. 1), combining it with hyperpolarization and automa-tion allowed for convenient monitoring of hydrogenation reactions, spin order manipulation, and preparation of hyperpolarized agents for MRI. To make use of $^1$H polarization generated by pH$_2$, we imple-mented an out-of-phase spin echo (OPE) sequence (see methods)[58,59], which has recently been used for gas imaging[60]. The sequence tailored for inhomogeneous magnetic fields allowed us to increase the signal-to-noise ratio (SNR) by a factor of five (Fig. 2).

In addition, we modified the OPE sequence such that it allowed us to spy on the spin evolution during SOT in real time without losing hyperpolarization and to reduce the sample consumption (Fig. 3).

We used this approach to study the hydrogenation of vinyl acetate (VA) and vinyl pyruvate (VP), two popular precursors for PHIP-SAH. Such optimizations are essential for moving towards in vivo applica-tions. We demonstrated the potential of our automated PHIP polarizer using 1-$^{13}$C-ethyl pyruvate-$d6$ (1-$^{13}$C-EP-$d6$), a highly relevant molecule that can be used as a metabolic tracer as is[61] or as pyruvate after cleaving the ester[21]. We achieved up to ~7% $^{13}$C polarization using ESOTHERIC SOT[53] (efficient spin order transfer to heteronuclei via relayed inept chains) and used the signal to acquire high-resolution $^{13}$C images of the reactor in situ and a 3D printed phantom.

## Results and discussion

### Automated half-tesla pH$_2$-polarizer
We built a gas-liquid control system around a portable teaching MRI system with a 0.55 T permanent magnet and a 10 mm bore. This system operated seven solenoid gas valves, two high-performance liquid chromatography valves, a syringe pump, and a radio-frequency (RF) unit. For further details on the system, see methods.

The custom-built gas-liquid path enabled automatic sample loading, disposal, hyperpolarization and quantification. Furthermore, the design of the system allowed us to perform multiple experiments in a row without any human intervention at all, e.g., for extensive parameter variation. For example, a single load of 20 mL allowed us to perform and vary the parameters in 60 hyperpolarization experiments within one hour, resulting in a duty cycle of only one minute.

### PHIP at the inhomogeneous 0.55 T magnetic field
Naturally, hydrogens transferred from pH$_2$ to a C=C double bond are hyperpolarized in PHIP. When these spins are weakly coupled (their

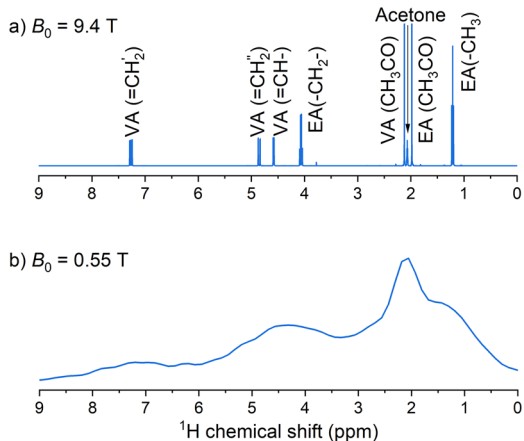

**Fig. 1 | Comparison between high-field, high-resolution, and low-field, low-resolution NMR spectra.** [1]H-NMR spectra of 50 mM vinyl acetate (VA) and 50 mM ethyl acetate (EA) dissolved in acetone-*d6* measured with a 9.4 T high-resolution NMR spectrometer (**a**) and the 0.55 T MRI (**b**). The inhomogeneity of the magnetic field and its limited strength render detailed chemical analysis impossible.

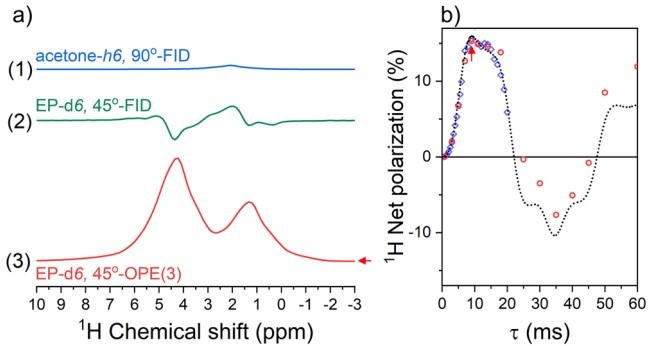

**Fig. 2 | Ethyl pyruvate (EP) hyperpolarized at 0.55 T. a** Conversion of the hyperpolarized anti-phase EP-*d6* spectrum (a-2) into an in-phase spectrum (a-3) boosted the SNR about six times. **b** Experimental (red circles) and simulated (dashed line) [1]H polarization achieved by the out-of-phase spin order transfer sequence with 3 refocusing elements (45°-OPE(3)) as a function of refocusing interval $\tau$. The spectrum for the highest polarization at $\tau = 10$ ms (arrow) is shown in a-3. Note that individual OPE-45 (red indicators) and longitudinal 45°-CPMG experiments (blue indicators) yielded very similar results (see Fig. 3 for details on 45°-CPMG SOT). The in-phase spectrum (a-3) was easier to analyze and less affected by the field inhomogeneity than the PASADENA spectrum (a-2). The estimated polarization of each proton was 14.4% (Supplementary Fig. 4), about one third of the theoretical maximum of 50% for a two-spin system[65]. Note that a new 300 µL sample injection was required for each red data point on (**b**). Note that spectrum of neat (~13.5 M), thermally polarized acetone-*h6*, acquired with a 90° excitation and acquisition sequence (90°-FID), showed a much lower amplitude as the signal of 50 mM hyperpolarized EP-*d6*. Spectra a-2,3 were obtained after bubbling 5 bar, 92% pH₂ for 45 s through the sample containing [Rh] = 3 mM, [vinyl pyruvate-*d6*] = 50 mM using 45°-FID (a-2) or 45°-OPE(3) with $\tau = 9$ ms (a-3). The simulated net polarization of EP-*d6* (**b**) was multiplied by exp(-$\tau$/60 ms) to accommodate signal decay phenomenologically (black dashed line).

chemical shift difference is much larger than their mutual scalar spin-spin interaction: $\delta\nu \gg J$), the conditions for pH₂ and synthesis allow dramatically enhanced nuclear alignment (PASADENA) are met[7]. We conducted this experiment inside the MRI system (45° excitation and acquisition) and observed a pair of anti-phase PASADENA doublets that were inhomogeneously broadened (Fig. 2a-2, see also methods). Because the signals have anti-phase shapes, partial line cancellation occurred in inhomogeneous $B_0$.

To date, dozens of sequences have been proposed to transfer the polarization from pH₂ to another nucleus[59,62–64]. Usually, SOT sequences with adiabatic or selective pulses allow the selected nucleus to approach the theoretically maximum polarization transfer. In our case, however, the magnetic field drifted about 100 Hz (or 5 ppm) per hour for [1]H (Supplementary Fig. 1), making such sequences impractical. Therefore we opted for the nonselective out-of-phase echo (OPE) SOT (methods)[58,65]. This way, the anti-phase PASADENA spectrum was converted into an in-phase spectrum providing 14.4% average [1]H polarization for the protons of ethyl pyruvate-*d6* (EP-*d6*) (Fig. 2a) and 16.2% for ethyl acetate-*d6* (EA-*d6*, Supplementary Fig. 4). The SNR of the in-phase spectrum was 5–6 times higher compared to the PASADENA spectrum.

To hydrogenate 50 mM of the precursor, we needed about 30 s. This period is longer than reported before (5 s)[33] and partially explains the lower-than-expected polarization. The use of higher temperatures, a higher pH₂ flow or more advanced pH₂ mixing may improve the matter[33,36,66].

## Effects of low magnetic field on spin order transfer

When the magnetic field $B_0$ is inhomogeneous, it is essential to refocus chemical shift evolution as rapidly as possible to prevent diffusion (or convection) induced dephasing of transverse magnetization[67–69]. Such a refocusing procedure typically consists of a 180° pulse surrounded by two $\tau$-intervals (see methods for details), which can be repeated $n$ times. The magnetic field can be inhomogeneous either on purpose (e.g., to prevent radiation damping[47,70]), or due to hardware imperfections; the latter was the case here.

At high magnetic fields, when the system of two spins is weakly coupled, the kinetics of polarization transfer (from anti-phase PASADENA spin order to in-phase spin order after the initial 45° excitation) is given by $\sin(2\pi J\tau n)$, and hence the optimal $\tau$-interval is given by $\tau = 1/4Jn$[65]. For a weakly coupled system, the total duration needed to transfer the most spin order into in-phase magnetization is constant: $n\tau = 1/4J$. So, for $J = 7.12$ Hz (corresponds to EP-*d6*), the first maximum for the SOT is achieved for $n\tau = 35.1$ ms. However, this is not the case for systems in the intermediate coupling regime or when $\delta\nu \sim J$. Note that the numerically calculated polarization transfer in this regime does not have a pure sine form (Fig. 2b) as expected for weakly coupled systems.

All the hyperpolarized protons investigated here fall into the intermediate coupling regime (EC, EA-*d6*, and EP-*d6*, see coupling constants in methods). More generally, a chemical shift difference of 1 ppm corresponds to 24 Hz for [1]H, and the typical [1]H chemical shift difference is below 5 ppm with $J \sim 10$ Hz, meaning that all protons at the $B_0 \sim 0.55$ T are at least approaching intermediate coupling conditions.

## "Spying" on nuclear spins

A consequence of the intermediate coupling between the protons[71] is the non-sinusoidal polarization transfer (Fig. 2b), which also depends non-linearly on the number of refocusing pulses (Fig. 3a, b).

To investigate (and observe) this spin evolution experimentally, we would need hundreds of 45°-OPE experiments and hundreds of samples: at least one sample for each number of echoes (100) and 20 different $\tau$-intervals, resulting in a total of 2000 samples to reproduce the simulated polarization transfer 2D map in Fig. 3.

As such numbers are unrealistic, we propose to use a different approach here. Instead of repeating one full experiment for each echo time, we suggest saving each echo of a full, CPMG-like (Carr–Purcell–Meiboom–Gill)[68] echo train instead (for each $\tau$).

In CPMG experiments, spin echoes appear in between the refocussing pulses spaced by $2\tau$. Thus, we set the middle of the acquisition window to the middle of the expected echo and measured nine data points with a 25 kHz sampling rate. Although the amplitudes

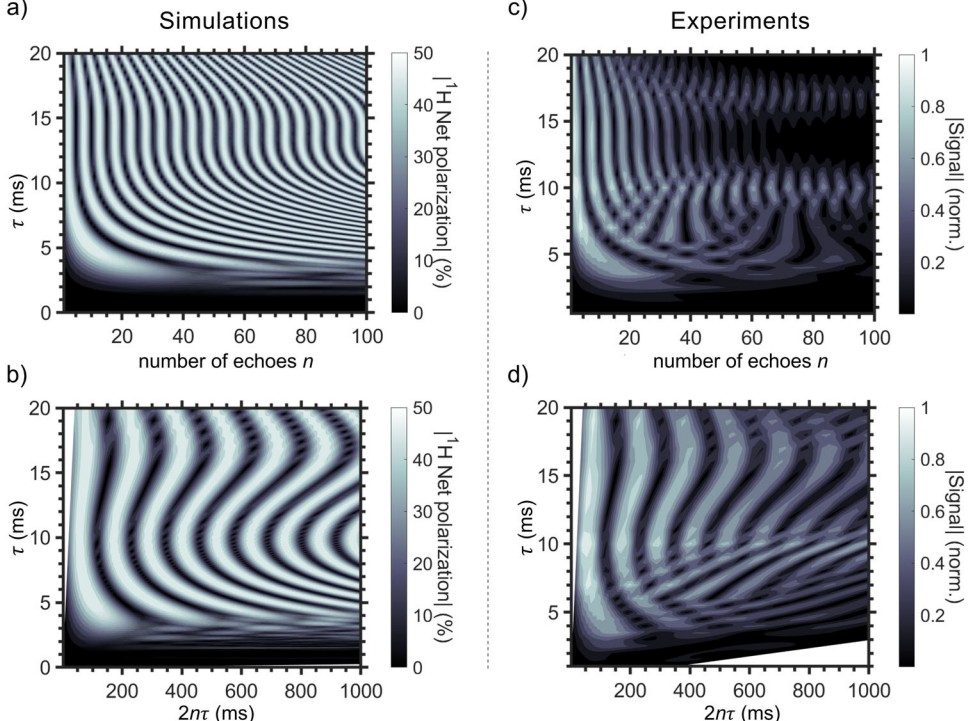

**Fig. 3 | Highly accelerated observation of spin order transfer in EP-*d6* during PHIP at 0.55 T.** In-phase PHIP signal simulated for 45° out-of-phase spin echo sequence with *n* refocusing elements and *τ* intervals (45°-sOPE(*n*,*τ*), **a**, **b**) and experimentally measured signals (**c**, **d**) using a 45°-CPMG sequence as a function of *τ* and a number of echoes *n* (**a**, **c**), or as a function of *τ* and 2*nτ* (**b**, **d**). The simulations were well reproduced by the experiments and helped to identify parameters suited for the spin order transfer. Here, each line in the polarization transfer map (**c**) was measured using one sample, while 100 would be necessary with conventional 45°-OPE. Simulations: only two protons were considered;

chemical shifts and *J*-coupling constants are given in methods; relaxation and magnetic field inhomogeneity were not considered; duration of the 90° and 180° pulses were 30 μs and 60 μs, respectively. Experiments: samples were prepared by mixing [Rh] = 3 mM with 50 mM of vinyl pyruvate-*d6* (VP-*d6*) in acetone-*d6*; the pH₂ pressure was 15 bar; the signal decayed due to relaxation, imperfect refocusing RF pulses, diffusion, and oscillation of the external magnetic field; the durations of the 90° and 180° pulses were 33 μs and 66 μs, respectively. Analogous maps were measured for EA-*d6* and EC (Supplementary Figs. 2 and 3). The 45°-CPMG values shown in Fig. 2b are taken from (**c**) for *n* = 3.

of each point were very similar, the absolute maximum point was attributed to the signal of the echo.

By measuring the amplitude of the echo, we lose spectral resolution. However, we can instead follow (or spy on) the nuclear spin evolution over many echo intervals. This way, only one, 300 μL injection of the sample was needed to measure one line of the 2D spin evolution map (Fig. 3c, d, and Supplementary Figs. 2, 3). Using the 45°-OPE approach, we would need one sample injection per data point (or 30 mL of the sample) to follow the spin evolution over the course of 100 spin echoes. Instead, a single injection of 300 μL was sufficient to acquire 100 echoes with 45°-CPMG. With the duty cycle of 1 min per sample, the entire SOT map (Fig. 3c) was measured in less than half an hour. Both methods gave comparable results (Fig. 2b) and were in agreement with the simulations (Fig. 3).

As we used deuterated solvents, the echo was dominated by the signal from the hyperpolarized spins. Other signal-suppressing pulse sequence elements[72,73] may help reduce other contributions, too.

**Implications at inhomogeneous fields (fast T₂* decay)**
We did not consider the relaxation effects in the simulations, although they play an important role in the experiments as the signal decays with an increasing number of echoes and time (Figs. 2b and 3c, d). T₂ relaxation, sample size, diffusion, and homogeneity of $B_0$ and $B_1$ fields contribute to signal decay. However, we had to consider another factor, i.e., the rapid oscillations of the magnetic field, which is uncommon for high-resolution superconducting NMR and MRI machines or actively locked magnets.

Permanent magnets are known for their limited homogeneity and field stability. In our case, the leading cause of the magnetic field oscillations of the $B_0$ field was temperature fluctuation. For our system, the standard deviation for the ¹H Larmor precession frequency of acetone-*h6* was 80 Hz (Supplementary Fig. 1), corresponding to a temperature variation of the permanent magnet of about 3–4 mK. We did not try to improve the temperature stability of the system further.

To investigate, if the polarizer is stable enough for SOT we measured the T₂ relaxation using the 90°-CPMG sequence as a function of the spin echo interval (Fig. 4). According to the Hahn equation[67–69], the signal decay in a CPMG experiment is given by the following equation:

$$M_{tr} = M_0 \exp[-R_2^{obs} 2n\tau] :=$$
$$= M_0 \exp\left[-\left(\frac{1}{T_2} + D^*(2\tau)^2\right)2n\tau\right] \quad (1)$$

where $R_2^{obs}$ is an observed decay rate, $D^* = \gamma^2 G^2 D/12$ is the effective diffusion coefficient and, in our case, also reflects the stability of the $B_0$ magnetic field, $\gamma$ is a gyromagnetic ratio, $G$ is a magnetic field gradient in the system ($B_0$ field inhomogeneity), and $D^*$ is a diffusion coefficient (or convection). In the absence of diffusion, for a stable and homogeneous $B_0$ field, the observed rate of signal decay, $R_2^{obs}$, should not depend on the *τ*-interval ($D^* = 0$). However, as illustrated (Fig. 4), this was not the case here. For echo intervals *τ* > 170 ms, the observed $R_2^{obs}$ deviated from the $\frac{1}{T_2} + D^*(2\tau)^2$ dependence. The resulting fitted parameters were $T_2 = 3.81 \pm 0.01$ s and $D^* = 13.4 \pm 0.2$ s⁻¹. The Hahn Eq. (1) was valid only approximately when 2*τ* < $T_2$/10. A plausible explanation is that Eq. (1) is valid for inhomogeneous, but static magnetic fields,

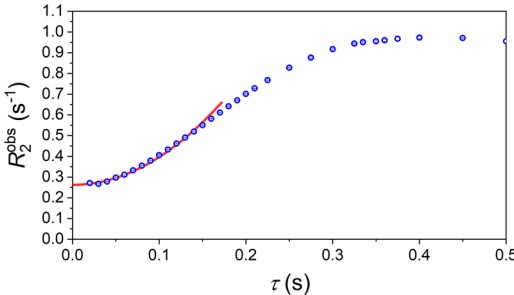

**Fig. 4 | Observed $T_2$ relaxation decay rate $R_2^{obs}$ as a function of $\tau$ (echo time is $2\tau$).** For each $\tau$, 169 echoes were acquired in a single $90°$-CPMG sequence. A monoexponential decay function fitted to the data was used to determine $R_2^{obs}$ for each $\tau$ (Eq. 1, circles with whiskers indicating standard deviation). These transversal relaxation rates were fitted with $R_2^{obs} = \frac{1}{T_2} + D^*(2\tau)^2$ for $\tau \leq 170$ ms (red line), resulting in $\frac{1}{T_2} = (0.262 \pm 0.001)$ s$^{-1}$ or $T_2 = (3.81 \pm 0.01)$ s and $D^* = (13.4 \pm 0.2)$ s$^{-1}$. Sample: the 10 mm tube with 300 μL of acetone-$h6$; the entire sample was inside the $B_1$ coil. Each measurement was repeated three times, after which the results were averaged (symbols).

which is not the case in our experiments. One reason for this deviation can be that the magnetic drift (not fluctuations) starts playing a more significant role for longer $\tau$-intervals. Note that for all $\tau$, which were below 20 ms in the $45°$-CPMG experiment (Fig. 3), the observed relaxation times did not deviate much from the intrinsic $T_2$ (Fig. 4). These measurements show that our system is stable enough for efficient polarization transfer.

**Reaction monitoring at an inhomogeneous low magnetic field**
High-resolution NMR allows us to observe individual chemical species and their transformations. This however appears to be impossible to achieve with the low and inhomogeneous magnetic fields used here (Fig. 1) due to their limited chemical shift resolution and low sensitivity. Despite this, since our product is spin-labeled by pH$_2$, we can still determine the chemical reaction rate. The hydrogenation kinetics of the precursor V to the product VHH is as follows:

$$V \xrightarrow{k_1 = [H_2][Rh]k_3} VHH, \qquad (2)$$

where $k_3$ is the trimolecular reaction rate constant, $k_1 = [H_2][Rh]k_3$ is a pseudo-first-order rate constant because one can assume that the concentrations of the catalyst [Rh] and hydrogen $[H_2]$ are constant during the entire hydrogenation experiment.

Because hyperpolarization is transient, the hyperpolarized signal decays (i.e., relaxes to equilibrium), which can be expressed phenomenologically in the same way as a chemical reaction:

$$VHH^* \xrightarrow{R} 0 \qquad (3)$$

Here VHH$^*$ is the product of concentration and polarization. As a result of the hydrogenation, the product gains polarization $P_0$. Under these assumptions, the evolution of VHH$^*$ can be derived as a function of bubbling time (for details see Supplementary information of ref. [27]):

$$[VHH^*] = \frac{[V_0]P_0}{1 - R/k_1}\left(e^{-\tau_b R} - e^{-\tau_b k_1}\right), \qquad (4)$$

where $\tau_b$ is the H$_2$ bubbling time and $[V_0]$ is the initial concentration of the precursor V. Hydrogenation kinetics for vinyl acetate to ethyl acetate (VA-$d6 \to$ EA-$d6$, Fig. 5a) and vinyl pyruvate to ethyl pyruvate(VP-$d6 \to$ EP-$d6$, Fig. 5b) were measured and fitted using Eq. (4).

The increase of pH$_2$ pressure from 5 bar to 15 bar resulted in an increased hydrogenation rate for VA-$d6 \to$ EP-$d6$ from

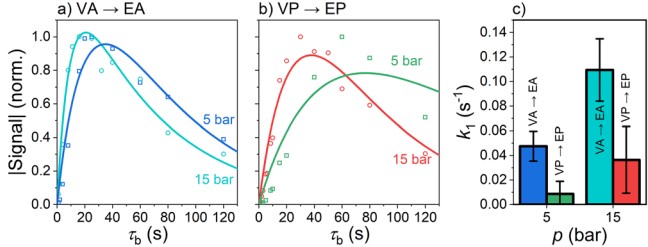

**Fig. 5 | Hydrogenation reaction kinetics for acetate and pyruvate precursors.** Normalized hyperpolarized signal of ethyl acetate-$d6$ (EA-$d6$, 5 bar – blue and 15 bar – cyan, **a**) and ethyl pyruvate-$d6$ (EP-$d6$, 5 bar – green and 15 bar – red, **b**) as a function of bubbling time $\tau_b$ for two pH$_2$ hydrogenation pressures. An increase in pH$_2$ pressure proportionally increased the hydrogenation rate; the hydrogenation of vinyl acetate (VA) was always faster than that of vinyl pyruvate (VP). The sample was prepared by mixing [Rh] = 3 mM with 50 mM of the hydrogenation precursor VA-$d6$ (**a**) and VP-$d6$ (**b**) in acetone-$d6$. The kinetics were fitted with two exponential decay functions of the form $A(e^{-\tau_b/T_1} - e^{-k_1\tau_b})$ with a shared $T_1 = 64 \pm 14$ s (here and below fit value and its standard errors are given). The reaction rate constants were measured to be: VA-$d6 \to$ EA-$d6$, $k_1(5\text{bar}) = 0.047 \pm 0.012$ s$^{-1}$, $k_1(15\text{bar}) = 0.11 \pm 0.025$ s$^{-1}$ and VP-$d6 \to$ EP-$d6$, $k_1(5\text{bar}) = 0.0086 \pm 0.01$ s$^{-1}$ and $k_1(15\text{bar}) = 0.036 \pm 0.027$ s$^{-1}$ for EP-$d6$. The bar plot (**c**) visualizes the difference between these rates; whiskers indicate the standard error.

$k_1 = 0.047 \pm 0.012$ s$^{-1}$ to $k_1 = 0.11 \pm 0.03$ s$^{-1}$. A further increase in pressure was not possible as the manufacturer's pressure limit for the used NMR tubes was 200 psi ($\approx$13.7 bar). The reaction can be accelerated further by using elevated temperatures. This option is not yet available in our setup; all presented experiments were carried out at room temperature (22 °C).

Under identical conditions, VP was hydrogenated considerably slower than VA (Fig. 5). It is known that the activity of the catalyst and hence the rate of hydrogenation depends on a number of parameters. The coordination of the solvent, the counter ion, the substrate, and the product can induce catalyst inhibition[74]. In the present case, the 1,2-dicarbonyl moiety of pyruvate (in VP and EP) likely coordinated with the Rh$^+$ ion, thereby decreasing the activity of the catalyst. Higher temperatures[33] or specially optimized catalysts are likely to accelerate the hydrogenation rate further.

In addition, the hydrogenation reaction may still be limited by the solubility of hydrogen in the solution to be hyperpolarized. The straightforward way to address this would be to supply more hydrogen, e.g., by using larger tubings or higher flow. In the literature, however, alternative methods were reported that should also be considered, including semipermeable membrane reactors, ideal for disturbing the magnetic field less and continuous polarization[36,75,76], as well as the ultrasonic spraying of the solution into a reaction chamber pressurized with pH$_2$[66].

Although there is much room for improvement in the chemical processes, it is advantageous that the NMR parameters such as chemical shifts and $^1$H-$^1$H $J$-coupling interactions are constant within the temperature range between 290 and 330 K (Supplementary Figs. 5–16), simplifying optimization of the SOT.

## $^{13}$C polarization and imaging
Using the same experimental conditions, ESOTHERIC SOT (see methods), and numerical simulations (Supplementary Fig. 17), we hyperpolarized 1-$^{13}$C-EP-$d6$ to ~7% $^{13}$C polarization with two chemical shift refocusing pulse loops (Fig. 6). Probing the decaying polarization yielded a lifetime of about 1 min. The design and automation of the setup featured a fast duty cycle of about 1 min, allowing quasi-continuous production of hyperpolarized tracers. Note that the magnetization (or signal intensity) of 50 mM $^{13}$C tracer hyperpolarized to 7% is approximately equal to ~6000 M thermally polarized tracer at 0.55 T (Fig. 6a). This exceeds the concentration of water (~55 M)

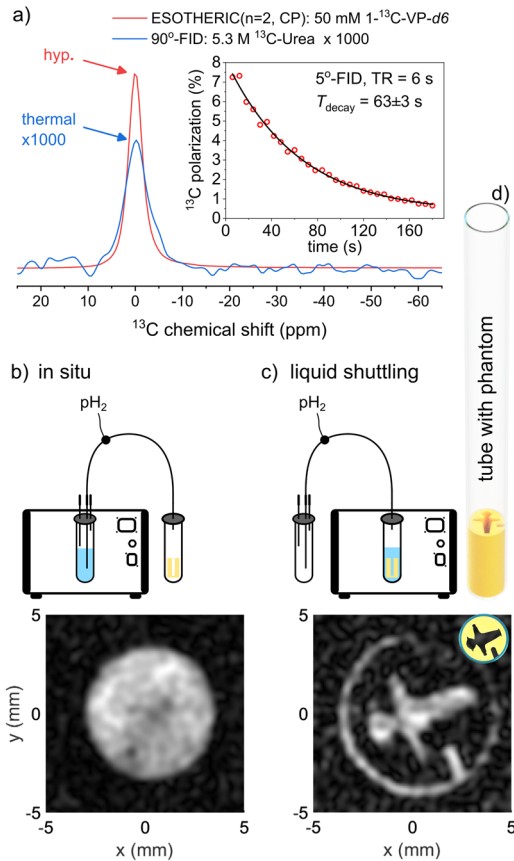

**Fig. 6 | $^{13}C$ Hyperpolarization and imaging of 1-$^{13}C$-pyruvate.** Spectroscopy (a) and imaging (b) of hyperpolarized 1–$^{13}C$-ethyl pyruvate-*d6* (1-$^{13}C$-EP-*d6*) Strongly hyperpolarized $^{13}C$ NMR signal of 50 mM 1-$^{13}C$-EP-*d6* was observed (a, red) and quantified to -7% $^{13}C$ polarization with respect to a thermally polarized, 5.3 M $^{13}C$-urea sample (a, blue, magnified 1000 times, $\tau_b = 20$ s hydrogenation time, ESO-THERIC SOT: $n_1 = n_2 = n_3 = 2$, $\tau_1 = \tau_3 = 169$ ms, and $\tau_2 = 90$ ms, see methods and Supplementary Fig. 17). Monitoring the decaying hyperpolarized signal using 5° excitation and acquisition scheme (5°-FID) every 6 s yielded a monoexponential decay constant of $63 \pm 3$ s. The signal was strong enough to record ten consecutive FLASH images in 1.5 s each, either in situ (b) or after shuttling the liquid into a tube holding a 3D-printed phantom (c, d, 5° excitation, cartesian encoding without acceleration, matrix of 32 × 32, repetition time of 50 ms). Schemes in (b) and (c) demonstrate the imaging site and liquid shuttling. The black hole in image (b) is the position of the 1/32" capillary. Note that ten images were acquired before the signal vanished.

by more than 100-fold. At high magnetic fields, the same ESOTHERIC SOT achieved almost P($^{13}C$) = 60%[21], indicating that further optimization is possible (as is improved hydrogenation).

This level of polarization was sufficient for acquiring multiple, consecutive FLASH[77] images, both in situ, directly after the polarization, and after transferring the sample to a 3D printed phantom (Fig. 6b, c).

For the latter, we hyperpolarized 1-$^{13}C$-EP-*d6* as before. Then, we transferred the liquid from the reaction chamber into a second receiver tube using the remaining pressure (via a three-way valve, see methods). Once the liquid was settled in the receiver tube, we exchanged the reaction chamber and receiver tube and initiated image acquisition. The receiver tube, a 10 mm flat bottom NMR tube, contained a 3D printed negative of the red light pedestrian traffic symbol popular in Berlin, Germany, known as "rotes Ampelmännchen" (Fig. 6d).

In both cases, high-resolution, fully-sampled cartesian $^{13}C$-FLASH MRI was acquired within 1.5 s without any dedicated acceleration

techniques (voxel size $(312\,\mu m)^2$, 5° excitations, repetition time 50 ms, 10 mm slice). Hyperpolarized $^{13}C$ signal was visible on ten images acquired every ~5 s. Limitations of the gradient power prohibited us from using smaller voxels and larger matrices.

Most polarizers for hydrogenative $^{13}C$ PHIP were reported to operate at low field (2–50 mT)[40,41,43,46,78], high field (7–9.4 T)[21,33,53,79] or using field cycling (μT–T); very few used the intermediate regime reported here (e.g., by ref. [70]). With these devices, $^{13}C$ polarizations up to several tens of percents were achieved, although there was significant variation in yield, tracers, and concentration. Amongst the latest approaches for PHIP-SAH, Mammone et al. polarized 100 μL of 55 mM $^{13}C_2$-cis-cinnamyl pyruvate ester-*d2* in a 22.6 mT electromagnet to 24% of $^{13}C$ polarization[44], and Marshall et al. polarized $^{13}C$-cis-cinnamyl pyruvate ester-*d* at 50 μT magnetic shield to 9.8% of $^{13}C$ polarization[80]. These polarizations are higher than those achieved here. We attribute this finding mostly to the inhomogeneous magnetic field of the MRI system (which was not designed for NMR). The advantages of our system include a high degree of automation, a short duty cycle, and the ability to run parameter variation experiments without human intervention. A comprehensive review on polarizers for hydrogenative PHIP was published recently[45]. It should be noted that very promising approaches to polarize biomolecules with SABRE, including pyruvate, are emerging at this moment[81,82].

While the volume of the reactor in this study was limited, the manufacturer (Pure Devices GmbH) can provide magnets with a bore's inner diameter of 20 mm, increasing the hydrogenation volume from 300 μL to more than 3 mL–comparable with preclinical dDNP systems[83,84]. About 30 mL of hyperpolarized solution with about 50% polarization is needed for human applications[2,3]. In this case, even larger reaction chambers (and magnetic systems) would be necessary; an alternative may be the quasi-continuous production and administration of hyperpolarized tracers as demonstrated here.

We expect that the automation and the short, 1-min duty cycle presented here will significantly benefit the translation of hyperpolarization in biochemical studies and medicine.

## Methods

### Chemicals
A homogeneous catalyst [1,4-Bis-(diphenylphosphino)-butan]-(1,5-cyclooctadien)-rhodium(I)-tetrafluoroborat ([Rh], 341134, Sigma-Aldrich), ethyl phenylpropiolate (EPh, E45309, Sigma-Aldrich), vinyl acetate (VA, 48486, Merck), vinyl pyruvate-*d6* (VP-*d6*) and 1-$^{13}C$-vinyl pyruvate-*d6* (1-$^{13}C$-VP-*d6*) synthesized according to our developed protocol[20], ethyl acetate-*d6* (EA-*d6*, 270989, Sigma-Aldrich), acetone-*d6* (444863, Sigma-Aldrich) were used here. The samples were prepared by mixing 50 mM of one of four precursors (EPh, VA-*d6*, VP-*d6*, 1-$^{13}C$-VP-*d6*) with 3 mM [Rh] in acetone-*d6*. The prepared sample was loaded into a 20 mL syringe and subsequently used in the PHIP experiment. The addition of $H_2$ in the presence of [Rh] resulted in the following chemical reactions: EPh → ethyl cinnamate (EC), VA-*d6* → ethyl acetate-*d6* (EA-*d6*), and VP-*d6* → ethyl pyruvate-*d6* (EP-*d6*) (Fig. 7).

$pH_2$ was prepared using an in-house built liquid-nitrogen-based parahydrogen generator (52% enrichment)[85] or with a generator based on a two-stage cryosystem[86] which was described elsewhere (92% enrichment).

### NMR and MRI systems
We used a high resolution, 9.4 T NMR system with a 5 mm broadband probe (400 MHz WB, NEO; BBFO, Bruker) and a benchtop, 0.55 T, 10 mm MRI system ("magspec" magnet unit, "drive L" console unit, Pure Devices GmbH). The manufacturer's software (TopSpin and open Matlab interface) and MestReNova v14.2.2 (Mestrelab Research) were used for signal acquisition and processing.

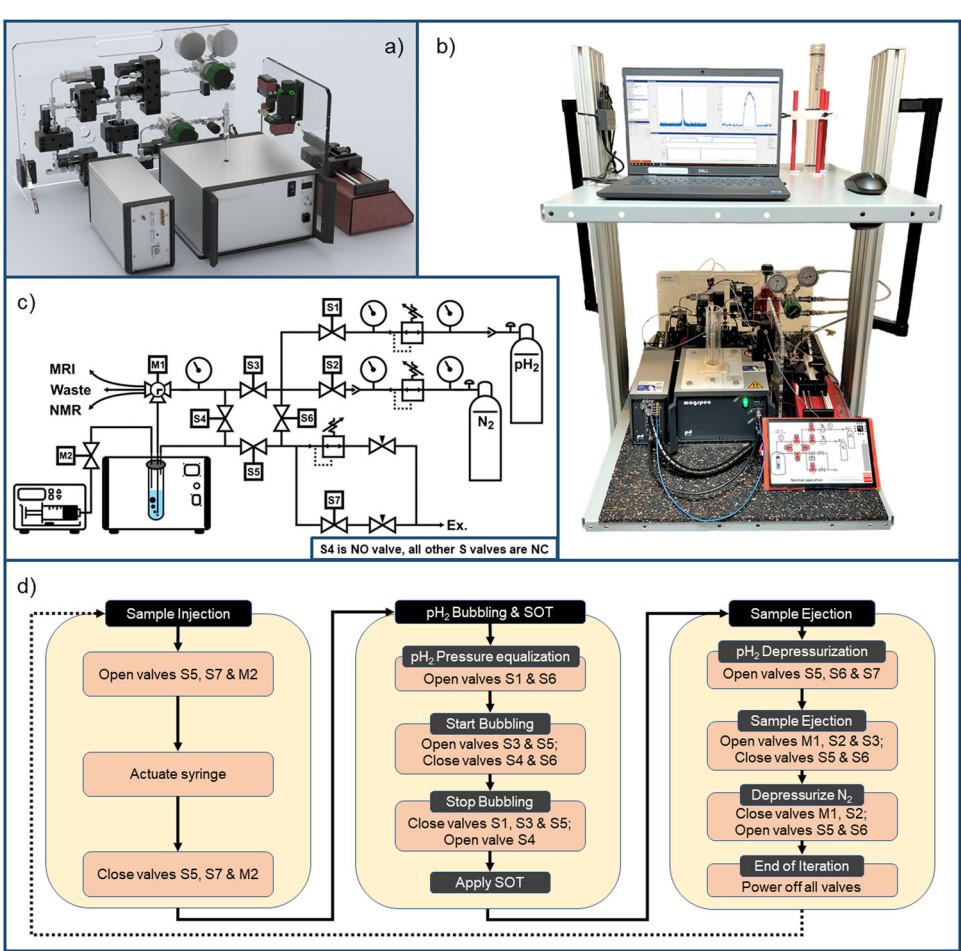

**Fig. 7 | Schematic of the chemical reactions used here to generate hyperpolarisation.** Hydrogenation of the three precursors used here: ethyl phenylpropiolate (EPh) → ethyl cinnamate (EC) (**a**), vinyl acetate-*d6* (VA-*d6*) → ethyl acetate-*d6* (EA-*d6*) (**b**), vinyl pyruvate-*d6* (VP-*d6*) → ethyl pyruvate-*d6* (EP-*d6*) (**c**). The relevant $^1H$ chemical shifts and $^1H$–$^1H$ J-coupling constants are indicated. pH$_2$ is marked in red.

## Hyperpolarization setup

The setup was built around the 0.55 T MRI system (Fig. 8a, b). A 10 mm high-pressure NMR tube served as a hydrogenation reactor and was placed in the MRI system (513-7PVH-7, Wilmad-LabGlass). A 3D-printed adapter was used to position the tube filled with 300 µL precursor-catalyst mixture in the isocenter (Tough PLA filament, S5 3D printer, Ultimaker).

## Precursor reservoir and injection

The precursor-catalyst solution was filled into a 20 mL syringe and placed into a syringe pump (BD Discardit II, Becton Dickinson, LA-110, Landgraf Laborsysteme HLL GmbH). The syringe was connected to the NMR tube using a 1/16" FEP tubing (1548 L, Idex), including a custom-built shut-off valve to prevent overpressure in the syringe (P-732, Idex). The pump was controlled with the "drive L" console. Typically, 300 µL were injected into the reactor.

## Hydrogenation and sample ejection

A second tube (1/32" PEEK, 1569, Idex) was used to either inject pH$_2$ into the reactor or to retrieve the sample from the reactor (for cleaning or secondary use of the sample in another NMR/MRI system). The tube

**Fig. 8 | The portable polarizer used in this study.** Rendering (**a**), photo (**b**), and diagram (**c**) of the polarizer presented here, and a typical automated hyperpolarization protocol (**d**). The hyperpolarization routine consisted of (1) cleaning by flushing the gas lines and tube with N$_2$ for 5 s; (2) releasing N$_2$ pressure from all gas lines; (3) filling the reactor (10 mm tube) with precursor solution from the syringe, (4) pressurization of the reactor with pH$_2$ and bubbling for $\tau_b$, (5) termination of the gas supply and equilibration of the pressure in the NMR tube for 1 s; (6) application of spin order transfer (SOT) sequence and (optional) MR signal acquisition, and (7) flushing out the sample using N$_2$ pressure. If desired, the entire cycle was repeated several times, e.g., while varying a given parameter. 20 mL of precursor–catalyst solution was sufficient for 60 experiments with a duty cycle of 1 min, which were carried out automatically. Any SOT can be used; 45°-FID, 45°-OPE, and 45°,90°-CPMG and ESOTHERIC[53] sequences (Fig. 9) were tested here.

was fed through a hole drilled through the cap of the NMR tube (sealed with epoxy, Loctite 3090, Henkel) and cut to match the length of the tube.

To inject $pH_2$ and eject the sample, a 4-way valve was used in a 3-way configuration (Z-UC-V-101L, Idex). Here, the common port was connected to the NMR tube, and the others with $pH_2$ supply and outlet. This allowed for a sample ejection by operating the valve connecting the NMR tube to the polarizer's sample outlet. So, a single PEEK tube serves as $pH_2$ supply and as an ejection path.

This fluid path (Fig. 8c) enabled automatic sample loading, performing hyperpolarization experiments, and in situ quantification. With the sample disposal being automated as well, multiple experiments can be performed in a row without a user's intervention (e.g., for parameter variation experiments).

### Custom-automated high-pressure PEEK valves
The manual shut-off valve and 4-way switching valve were actuated by servo motors (DS8020, MASTER) equipped with a custom-made 3D-printed holder. A lab power supply provided the respective power for the servo motors (BT-305, BASETech). The servo motors were controlled using a microcontroller programmed with custom firmware (Arduino framework, ATmega328 chipset).

### Gas supply unit
Stainless-steel tubes (1/8") were used to connect the valves (24 V, A52301002.012NO, A52301002.032XX, GSR Ventiltechnik GmbH & Co. KG), gauges, and other parts of the gas supply unit, mounted on a laser-cut PMMA sheet (8 mm Plexiglas XT, Röhm GmbH; Speedy 360, Trotec Laser GmbH, Fig. 8).

We operated the system with $pH_2$ at 5 or 15 bar. $N_2$ at 5 bar was used only for sample shuttling and system flushing. All components were rated to 30 bar or more.

A newly developed, general-purpose, microcontroller-driven control board was used to operate the gas supply unit. A microcontroller with custom firmware operated all valves according to a programmed valve switching schedule (ESP32, Espressif). The control board featured two power supplies (3.3 V and 5.0 V) for the microcontroller, the sensors, and the touch display (TSR_1-2450, TSR_1-2433, Traco Electronic AG). The touch display was used as a user interface and flashed with custom firmware (NX1060P101-011C-I, Nextion). The digital outputs of the microcontroller were used to switch N-channel MOSFETs, connected to a 24 V switch-mode power supply (GST60A24-P1J, Meanwell) to drive the solenoid gas valves (DMG3402L-7, Diodes Incorporated). The control board also featured a hydrogen sensor (MQ-8, Hanwei Electronics Co., Ltd) with a nominal sensitivity of 100 ppm. The sensor shuts off the $H_2$ supply if hydrogen is detected in the atmosphere.

### Hyperpolarization procedure
The polarization procedure included loading the precursor with the catalyst, injecting $pH_2$, and commencing the hydrogenation, SOT, signal acquisition, and sample ejection and required accurate timing of each step (Fig. 8d). The procedure was programmed into the MRI console (Matlab, drive L, Pure Devices GmbH), whose digital outputs were used to trigger the desired event (TTL pulses to the syringe pump and gas supply unit).

### Signal acquisition
For the SOT and signal acquisition, four different pulse sequences were implemented and used (Fig. 9): 45°-OPE, φ-CPMG, and ESOTHERIC[53]. The parameters for ESOTHERIC[53] SOT: $n_1$, $n_2$, $n_3$, $\tau_1$, $\tau_2$ and $\tau_3$ were varied and optimized numerically for 1-$^{13}$C-EP-$d6$ spin system (Supplementary Fig. 17).

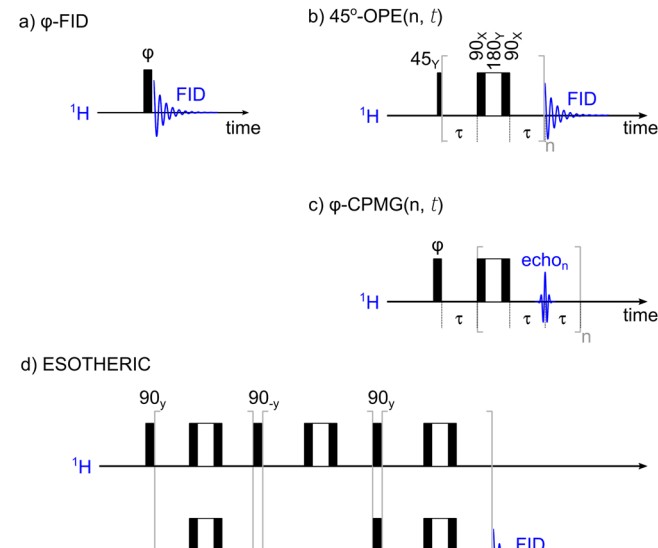

**Fig. 9 | Schematic of the NMR sequences used in this study.** These sequences are φ-FID (**a**), 45°-out-of-phase echo (OPE) (**b**), φ-CPMG (**c**), and ESOTHERIC[53] (**d**). In the case of thermal polarization, φ = 90° results in maximum polarization (in **a** and **c**), while in the case of PHIP (PASADENA[7]), φ = 45° should be used instead. Note that composite refocusing pulses ($90^{\circ}_x 180^{\circ}_y 90^{\circ}_x$) were used, and the difference between (**b**) and (**c**) is the signal acquisition (FID or multiple echoes. FID: free induction decay, OPE: out-of-phase echo, CPMG: Carr-Purcell-Meiboom-Gill sequence, $n$, $n_1$, $n_2$, and $n_3$ are the number of refocusing pulses, ESOTHERIC: efficient spin order transfer to heteronuclei via relayed inept chains.

## Data availability
All data generated in this study together with processed data and short descriptions are available together with the manuscript (Source-data.zip). Source data are provided with this paper.

## Code availability
Spin simulation scripts, together with scripts used for data acquisition and processing, are available together with the manuscript (Source-code.zip, Matlab).

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

## Acknowledgements

We acknowledge funding from the German Federal Ministry of Education and Research (BMBF) within the framework of the e:Med research and funding concept (01ZX1915C), DFG (PR 1868/3-1, HO-4604/2-2, HO-4604/3, GRK2154-2019, EXC2167, FOR5042, TRR287), DAAD-RISE program. MOIN CC was founded by a grant from the European Regional Development Fund (ERDF) and the Zukunftsprogramm Wirtschaft of Schleswig-Holstein (Project no. 122-09-053). Pure Devices GmbH team, Toni Drießle and Stefan Wintzheimer, in particular, for the assistance with the benchtop MRI. A.N.P. thanks George J. Hirasaki and Philip Singer for stimulating the discussion. A.S. was participating in this research during his internship in Kiel [a] in 2022 supported by the DAAD-RISE program.

## Author contributions

A.N.P., J.B.H.—design of the project; A.N.P., F.E.—collected the data; F.E., A.S.—design and assembly of the equipment; A.N.P.—data analysis, spin dynamic analysis; A.S., A.N.P.—data analysis tools; A.B., R.H.—synthesis of VP-d6; C.A.—assistance with experiments, high field NMR; A.N.P., F.E.—original draft; A.N.P., J.B.H.—funding acquisition. All authors have contributed to the preparation of the manuscript and approved the final version of the manuscript.

## Funding

## Competing interests

The authors declare no competing interests.
