## [Peer Review File · Nature Communications]

REVIEWER COMMENTS

Reviewer #1 (Remarks to the Author):

This is a very interesting paper about an apparatus design for producing hyperpolarized, PHIP-based reagents. It is compact, uses a relatively simple magnet, and could probably be duplicated reasonably by many labs. It also nicely takes advantage of the speed associated with the PHIP process. As such, this would be a great paper for Review of Scientific Instruments, or perhaps for the Journal of Magnetic Resonance, and this reviewer would be enthusiastic.

Unfortunately, for this journal (which intends to publish papers with significant novelty) it falls far short. Compensation for the effects of a drifting magnet is well known; none of the components presented here are new; and none of the results are particularly novel. Again, putting them together into a relatively inexpensive and reproducible package is a nice result-the paper just does not belong in a high impact journal.

Reviewer #2 (Remarks to the Author):

The authors present methods for hyperpolarization (PHIP) and monitoring of spin state evolution using a commercially available benchtop MRI system. The novel aspects of the study are: repurposing a "teaching" MRI system as a hyperpolarizer, automation, and a pulse sequence to monitor the spin evolution. I think the study is conceptually interesting and appealing, but the manuscript requires clarification and additional experiments.

Overall

1. Results are mostly anecdotal without comparisons to standard methods or detailed investigations.
2. The manuscript is written with some "assumed knowledge" of the reader and could be improved for a more general audience with additional clarification and contextualization.

Introduction

3. Although 0.55T is "low field" for an MRI system, 0.55T is on the higher end of field strengths used for PHIP polarizers, therefore the terminology is confusing throughout. Please provide more details of other purpose-built PHIP polarizers (e.g. field strength and magnet design) to help readers appreciate this, and please provide more context into the "high" 0.55T field strength.
4. Note that the Ivanov group has characterized SABRE across a range of field strengths from ultralow field to ultrahigh field. Please include relevant references
5. The advantages of using the 0.55T MRI system are not well described in the Introduction. The accessibility/availability of this benchtop system is clear, and I believe this is the primary advantage. Other than that, the authors state that it enables larger volumes – please clarify by how much and why it matters. The authors also state that it has built-in electronics for SOT and hardware for shimming – purpose-built PHIP polarizers also have these. Please clarify.
6. The introduction does not provide context for why the real-time "spying" is valuable and novel. Please describe how this compares to current workflows.

Results

7. No in-vivo results are provided. I think in vivo results are important to demonstrate the effectiveness of this technique for metabolic imaging. If this is not possible, even some phantom results showing ^{13}C imaging or spectroscopy would be helpful.
8. No comparisons to other PHIP polarizer systems are provided. It is difficult to assess the relative effectiveness of the technique without a direct comparison to other established methods.
9. The purpose of Figure 1 was unclear to me. I agree that the 0.55T system is not well-suited for resolving spectra, mostly because of the field inhomogeneity not the SNR, but resolving spectra is not the intended use of the 0.55T system in this work. The 0.55T is used for hyperpolarization while another system would be used for NMR or MRI. I do not think this figure provides useful data.
10. The magnet drift of 100Hz in 1 hour could presumably be corrected using frequent re-shimming (the authors have already stated that shimming hardware is available). Please demonstrate the added value of the OPE SOT method compared to frequent re-shimming or justify why this is not possible.
11. Hyperpolarization of 14.4-16.2% is low compared to published values up to 50% for dDNP - please comment.
12. In the section "Effects of low magnetic field on spin order transfer", the first paragraph is about

field homogeneity (not field strength), which seems better suited for "Implications at inhomogeneous fields". I recommend moving this content to improve readability.

13. Please provide a comparison between the 45-CPMG method and the 45-OPE to verify that your 45-CPMG method is performing equivalently. This method is not validated for the reader.

14. Field inhomogeneity – provide reference values of field homogeneity for your 0.55T system, compared to other polarizers, to help contextualize.

Typos

15. Typo: SOT defined twice

Reviewer #3 (Remarks to the Author):

Ellermann et al. present a compact automated polarizer to prepare hyperpolarized molecules using parahydrogen-induced polarization (PHIP) based on a low-cost MRI scanner. They convert the anti-phase signal of the singlet 1H spin pair into an in-phase signal using an echo with refocusing composite pulses to acquire the hyperpolarized signal with a better sensitivity despite the low homogeneity of the magnet. They report 1H polarizations on the order of 15%. Their automated setup allows them to record a large data set of signal intensity vs echo time and number of echoes, which compares well with spin dynamics simulation. They show that the contribution of field inhomogeneity to losses during the echo is not significant for the echo times that they typically use. Finally, they use their setup to study the kinetics of hydrogenation for acetate and pyruvate precursors.

The results are interesting and will certainly benefit the parahydrogen community. Indeed, as pointed out by the Authors, automation is an important step towards more repeatable and reproducible PHIP experiments.

A strong weakness of the paper is that the presented instrument and its performance are not compared with previously reported examples. If other polarizers are indeed mentioned in the introduction, a paragraph in the discussion or conclusion should be devoted to this comparison. What is the difference between the polarizer presented in Ellermann's paper compared to previously reported ones? How does the performance compare with those (eg: optimal 1H polarization, duty cycle, cost, footprint,...)? Are the measured rates in Figure 5 more accurate than when measured with the good old shake-and-run experiment? What are the perspectives for improving the hardware presented here? Do the Authors intend to implement strategies to transfer the polarization to 13C? This is after all the strongest driving force for PHIP in MRI. Finally, I am no expert on the topic but I suspect that other automated systems exist that are not cited (eg. <https://doi.org/10.1002/cphc.202100667>)

If the manuscript is overall well structured, it is not easy to read. First, the text is split into too many paragraphs, sometimes of a single sentence. This is disruptive to the flow and makes it hard to follow the ideas that the Authors try to convey. Second, several variables and concepts are only defined in the methods section. The reader can therefore not always understand the main message (not only the details) without jumping to the end of the paper. Finally, there are many small mistakes in the language and the figure captions. I've tried to list as many as possible in the detailed comments below.

I suggest publishing this manuscript only after these two points above (comparison with the literature and readability) have been improved.

REVIEWER COMMENTS

Our answers are written with blue color.

Modified text is highlighted with yellow marker.

Reviewer #1 (Remarks to the Author):

This is a very interesting paper about an apparatus design for producing hyperpolarized, PHIP-based reagents. It is compact, uses a relatively simple magnet, and could probably be duplicated reasonably by many labs. It also nicely takes advantage of the speed associated with the PHIP process. As such, this would be a great paper for Review of Scientific Instruments, or perhaps for the Journal of Magnetic Resonance, and this reviewer would be enthusiastic.

Unfortunately, for this journal (which intends to publish papers with significant novelty) it falls far short. Compensation for the effects of a drifting magnet is well known; none of the components presented here are new; and none of the results are particularly novel. Again, putting them together into a relatively inexpensive and reproducible package is a nice result-the paper just does not belong in a high impact journal.

Answer: Dear reviewer, thank you for your time. We see it differently. We think that with such systems hyperpolarization is more accessible to many labs: we demonstrate a sophisticated polarizer which polarizes samples consecutively at the push of a button enabling analytics and even imaging. We believe that the whole system is more than just the some of its parts that makes this contribution novel. A lack of published PHIP polarizers shows how challenging it is to have the underlying physics, chemistry, and engineering work cohesively to such a degree that an actual device can consecutively polarize and analyze samples with no human intervention. Moreover, the work we present not only demonstrates how we overcame the physics and engineering challenges, but also, for the first time to our knowledge, proves that PHIP-based polarizers can be designed and implemented with such a high degree of automation.

Hence, especially with the new imaging experiments which have been added to the manuscript, we see our work in a high impact journal. We are certain that this work will fit the broad audience of *Nature Communications*.

Reviewer #2 (Remarks to the Author):

The authors present methods for hyperpolarization (PHIP) and monitoring of spin state evolution using a commercially available benchtop MRI system. The novel aspects of the study are: repurposing a “teaching” MRI system as a hyperpolarizer, automation, and a pulse sequence to monitor the spin evolution. I think the study is conceptually interesting and appealing, but the manuscript requires clarification and additional experiments.

Overall

1. Results are mostly anecdotal without comparisons to standard methods or detailed investigations.

Answer:

We changed introduction, results and conclusions that gives now more context and comparison. Examples are given:

New text in introduction:

In the literature, several approaches for PHIP are described, from "shake and run" with an NMR tube³¹ to complex, stand-alone^{17,32} or in situ polarizers for MRI³³ or NMR^{22,26,34-36}.

The very first PHIP polarizers for MRI contrast agents relied on spin order transfer (SOT) from parahydrogen via strong spin-spin interactions between ¹H and ¹³C during MFC at fields below 0.1 mT^{37,38}. Soon after, polarizers with RF induced SOT were introduced³⁹ which doubled the experimentally achieved polarization reaching ~50% ¹³C polarization for partially deuterated 2-hydroxyethylpropionate, while only 21-25% was possible with MFC⁴⁰. Hyperpolarized 2-hydroxyethylpropionate is a non-active metabolic tracer and therefore has been used for MRI angiography instead³⁷. There are some more low-field systems with RF induced SOT which have been described⁴¹⁻⁴⁴.

In general, ultra-low (< 1 mT) and low-field (1 mT – 1 T) PHIP polarizers have the advantage of small physical size, low setup and running costs, and the fact that there is no need for cryogenics^{45,46}. So far, some of the highest polarization values with the hydrogenative PHIP-SAH approach, have been demonstrated using RF induced SOTs with superconducting high-field (~7 T) magnets^{21,47}.

The efficiency of SOT is limited if there are strong indirect interactions between different nuclei (¹H, ²H, ¹³C, etc.)^{48,49} and often 100% polarization is not accessible even theoretically with MFC^{40,50,51}. Ultra-low field conditions with MFC can be used for polarization of some molecules, however, in general they do not guarantee high polarization values and RF induced SOT is more flexible in this regard⁵²⁻⁵⁴. Therefore, we opted for the latter approach.

New text in results:

Higher temperatures³³ or specially optimized catalysts are required to accelerate the hydrogenation rate further.

In addition, the hydrogenation reaction is possibly still limited by the solubility rate of hydrogen in the solution to be hyperpolarized. The straightforward way to change it would be using thicker tubings or stronger flow. In the literature, however, alternative methods were reported that should also be considered such as: semipermeable membrane reactors ideal for low B_0 magnetic field disturbance and continuous polarization^{36,73,74}, as well as ultra-sonic spraying of the solution to be polarized into a reaction chamber pressurised with p H_2 ⁷⁵.

New text in conclusion:

Accelerated hydrogenation reactions of VA and VP at increased pressures were observed, and the reaction rates were evaluated. We needed ~30 s for a complete hydrogenation of 50 mM of the precursor, which is much longer than a recently reported 5 s hydrogenation time³³. The use of higher temperatures, a higher H₂ flow or more advanced H₂ mixing can improve it^{33,36,75}. The great advantage of a half-tesla benchtop is that it is (i) completely independent from the MRI or NMR system, (ii) portable with a footprint of 1 m², and (iii) after moving the setup to a new place it needs no more than a couple of minutes for automatic magnetic field calibration before usage. The manufacturer (Pure Devices GmbH) can provide magnets with a bore's inner diameter of 20 mm that will increase the hydrogenation volume from 300 μ L to more than 3 mL. This increase in volume will be already comparable with preclinical dDNP systems^{77,78}.

2. The manuscript is written with some “assumed knowledge” of the reader and could be improved for a more general audience with additional clarification and contextualization.

Answer:

I hope we clarified some missing concepts. Examples given below.

New text in results:

Effects of low magnetic field on spin order transfer. When the magnetic field B_0 is inhomogeneous, it is essential to refocus chemical shift evolution as rapidly as possible to prevent dephasing of transversal magnetization^{65–67}. Typical refocusing consists of two τ -intervals and 180° refocusing pulse in between (**Figure 9b-d**). This block can be repeated n times. The magnetic field can be inhomogeneous either on purpose (e.g., to prevent radiation damping^{49,68}) or due to imperfection of the hardware; the latter is the case here.

New text in results:

Alternatively, we propose measuring spin echoes with the 45°-CPMG sequence instead of the whole free induction decay as in the 45°-OPE (methods: **Figure 9b,c**). Both methods gave comparable results (**Figure 2b**).

In CPMG experiments, spin echoes appear each time after the τ delay from the last refocusing pulse. We set the middle of the acquisition window in the middle of the expected echo and measured 9 data points at a 25 kHz rate. Although the amplitudes were very similar, the absolute maximum point was attributed to the signal of the echo. By measuring the amplitude of the echo, we obviously do not have spectral resolution, however, we can instead follow (or “spy” on) the nuclear spin evolution over many echo intervals. Using this approach, we needed only one 300 μ L injection of the sample to measure one line of a 2D map (**Figure 3c,d**, and **SI, Figures S2, S3**). Using the 45°-OPE approach, we would need one sample injection per data point. As a consequence, we would need 30 mL of the sample to follow the spin evolution over the course of 100 spin-echoes.

3. Although 0.55T is “low field” for an MRI system, 0.55T is on the higher end of field strengths used for PHIP polarizers, therefore the terminology is confusing throughout. Please provide more details of other purpose-build PHIP polarizers (e.g. field strength and magnet design) to help readers appreciate this, and please provide more context into the “high” 0.55T field strength.

Answer:

We agree. By high-field systems we consider more of superconducting magnets, which are also very typical for PHIP polarization.

Example of new text in introduction:

In general, ultra-low (< 1 mT) and low-field (1 mT – 1 T) PHIP polarizers have the advantage of small physical size, low setup and running costs, and the fact that there is no need for cryogen^{45,46}. So far, some of the highest polarization values with the hydrogenative PHIP-SAH approach, have been demonstrated using RF induced SOTs with superconducting high-field (~7 T) magnets^{21,47}.

4. Note that the Ivanov group has characterized SABRE has across a range of field strengths from ultralow field to ultrahigh field. Please include relevant references

Answer:

Indeed, such systems can be repurposed to become a polarizer for MRI, however it was not built for this. We added now many references in the introduction, including the one mentioned above.

Example of new text in introduction:

In the literature, several approaches for PHIP are described, from "shake and run" with an NMR tube³¹ to complex, stand-alone^{17,32} or in situ polarizers for MRI³³ or NMR^{22,26,34–36}.

5. The advantages of using the 0.55T MRI system are not well described in the Introduction. The accessibility/availability of this benchtop system is clear, and I believe this is the primary advantage. Other than that, the authors state that it enables larger volumes – please clarify by how much and why it matters. The authors also state that it has built-in electronics for SOT and hardware for shimming – purpose-built PHIP polarizers also have these. Please clarify.

Answer:

Thank you for making that point.

New text in introduction:

This mobile MRI allows, by design, for a larger sample volume than NMRs, features all electronics for SOT, and incorporates sophisticated shimming hardware. The device neither requires cryogen, nor a certain power connection as it operates with a domestic power socket, hence, there are no particular installation requirements and no maintenance efforts to consider. We found it convenient to use a 300 μL sample size. Note that while larger volumes are possible, at high-resolution narrow bore NMR only 100 – 200 μL can be polarized due to spatial and B_1 homogeneity restrictions^{21,49}. The "teaching" MRI was convenient for automation of hyperpolarization processes and signal acquisition as it comes with all necessary RF and shimming electronics that lowers the barrier for the community to start using it. Moreover, it features digital input/output (IO) triggers which can be programmed within the NMR and MRI sequences. For example, an automatic B_0 field calibration was convenient when the system was moved to another lab site; calibration before use took only about 2 min. The mobility of the system makes it possible to place it wherever its necessary (e.g., in close proximity to a preclinical MRI).

New text in conclusion:

The great advantage of a half-tesla benchtop is that it is (i) completely independent from the MRI or NMR system, (ii) portable with a footprint of 1 m^2 , and (iii) after moving the setup to a new place it needs no more than a couple of minutes for automatic magnetic field calibration before usage. The manufacturer (Pure Devices GmbH) can provide magnets with a bore's inner diameter of 20 mm that will increase the hydrogenation

volume from 300 μL to more than 3 mL. This increase in volume will be already comparable with preclinical dDNP systems^{77,78}. About 30 mL of hyperpolarized solution with about 50% polarization is needed for human applications^{2,3}. In this case, even larger reaction chambers (and magnetic systems) would be necessary, or one can instead use semicontinuous automated production (and administration) of hyperpolarized small boluses demonstrated here.

6. The introduction does not provide context for why the real-time “spying” is valuable and novel. Please describe how this compares to current workflows.

Answer:

New text in introduction:

Our modifications of the OPE sequence enabled real-time spying on the spin evolution without the depletion of hyperpolarization. With this technique, we reduced consumption of the sample to be polarized by 100 times, since complete spin evolution of anti-phase to in-phase conversion was measured using one sample only.

Results

7. No in-vivo results are provided. I think in vivo results are important to demonstrate the effectiveness of this technique for metabolic imaging. If this is not possible, even some phantom results showing ^{13}C imaging or spectroscopy would be helpful.

Answer:

Now, the ^{13}C polarization and images of phantoms are provided in results and discussion section.

Text modified in many instances.

8. No comparisons to other PHIP polarizer systems are provided. It is difficult to assess the relative effectiveness of the technique without a direct comparison to other established methods.

New text in results and discussion:

In addition, the hydrogenation reaction is possibly still limited by the solubility rate of hydrogen in the solution to be hyperpolarized. The straightforward way to change it would be using thicker tubings or stronger flow. In the literature, however, alternative methods were reported that should also be considered such as: semipermeable membrane reactors ideal for low B_0 magnetic field disturbance and continuous polarization^{36,73,74}, as well as ultra-sonic spraying of the solution to be polarized into a reaction chamber pressurised with pH_2 ⁷⁵.

New text in conclusion:

The use of higher temperatures, a higher H_2 flow or more advanced H_2 mixing can improve it^{33,36,75}. The great advantage of a half-tesla benchtop is that it is (i) completely independent from the MRI or NMR system, (ii) portable with a footprint of 1 m^2 , and (iii) after moving the setup to a new place it needs no more than a couple of minutes for automatic magnetic field calibration before usage. The manufacturer (Pure Devices GmbH) can provide magnets with a bore's inner diameter of 20 mm that will increase the hydrogenation volume from 300 μL to more than 3 mL. This increase in volume will be already comparable with preclinical dDNP systems^{77,78}. About 30 mL of hyperpolarized solution with about 50% polarization is needed for human applications^{2,3}. In this case, even larger reaction chambers (and magnetic systems) would be necessary, or one can

instead use semicontinuous automated production (and administration) of hyperpolarized small boluses demonstrated here.

Using ESOTHERIC SOT, $1\text{-}^{13}\text{C-EP-d6}$ was hyperpolarized to ~7% and consequently imaged in the reactor in situ or after liquid shuttling to a phantom. With a duty cycle of only 1 min and a lifetime of polarization of 1 min, the system provides a semiconstant supply of ^{13}C hyperpolarization. Using magnetic field cycling, $1\text{-}^{13}\text{C-EP}$ was hyperpolarized to almost 12.3% ^{13}C polarization. At high magnetic fields, the same ESOTHERIC approach as we used here achieved almost 60%;²¹ indicating that the concept for RF SOT polarization is valid. However, further hydrogenation optimization can improve the results as already discussed above.

9. The purpose of Figure 1 was unclear to me. I agree that the 0.55T system is not well-suited for resolving spectra, mostly because of the field inhomogeneity not the SNR, but resolving spectra is not the intended use of the 0.55T system in this work. The 0.55T is used for hyperpolarization while another system would be used for NMR or MRI. I do not think this figure provides useful data.

Answer:

We think that this figure is useful to introduce low field NMR to a broader audience. A hydrogenation reaction can be studied without hyperpolarization if there is enough resolution, which is not our case, therefore, hyperpolarization is actually essential not only for the production of polarized agents for MRI but to study hydrogenation reactions in inhomogeneous low fields, too.

Modified text in introduction:

Although the low-field MRI system does not have a high spectral resolution (**Figure 1**), combined with hyperpolarization and automation, it allowed for convenient automated monitoring of hydrogenation reaction, spin order manipulation, and preparation of hyperpolarized agents for MRI.

10. The magnet drift of 100Hz in 1 hour could presumably be corrected using frequent re-shimming (the authors have already stated that shimming hardware is available). Please demonstrate the added value of the OPE SOT method compared to frequent re-shimming or justify why this is not possible.

Answer:

In the introduction we write:

To make use of ^1H polarization generated by pH_2 , we implemented an out-of-phase spin echo (OPE) sequence (see methods, **Figure 9**)^{57,58}, which has recently been used for gas imaging⁵⁹. This sequence is of particular interest at inhomogeneous magnetic fields, since it converts hyperpolarized anti-phase spectra to in-phase NMR spectra, yielding a substantial increase in sensitivity of hyperpolarized NMR.

Moreover, please have a look at the Figure 2a-2 and 2a-3.

OPE converts an antiphase spectrum into an in-phase spectrum. All spectra on 2a are acquired with the best possible shims within 2-3 minutes after previous shimming. So, there is no way to improve B_0 homogeneity. At the same time, you see the improved SNR when OPE is used instead of an FID.

We hope that the issue is clarified and the advantages and use of OPE is justified.

11. Hyperpolarization of 14.4-16.2% is low compared to published values up to 50% for dDNP - please comment.

Answer:

The conclusion was significantly extended, and this point was discussed there in the context of ^{13}C polarization:

Accelerated hydrogenation reactions of VA and VP at increased pressures were observed, and the reaction rates were evaluated. We needed ~ 30 s for a complete hydrogenation of 50 mM of the precursor, which is much longer than a recently reported 5 s hydrogenation time³³. The use of higher temperatures, a higher H_2 flow or more advanced H_2 mixing can improve it^{33,36,75}. The great advantage of a half-tesla benchtop is that it is (i) completely independent from the MRI or NMR system, (ii) portable with a footprint of 1 m², and (iii) after moving the setup to a new place it needs no more than a couple of minutes for automatic magnetic field calibration before usage. The manufacturer (Pure Devices GmbH) can provide magnets with a bore's inner diameter of 20 mm that will increase the hydrogenation volume from 300 μL to more than 3 mL. This increase in volume will be already comparable with preclinical dDNP systems^{77,78}. About 30 mL of hyperpolarized solution with about 50% polarization is needed for human applications^{2,3}. In this case, even larger reaction chambers (and magnetic systems) would be necessary, or one can instead use semicontinuous automated production (and administration) of hyperpolarized small boluses demonstrated here.

Using ESOTHERIC SOT, $1\text{-}^{13}\text{C}\text{-EP-}d_6$ was hyperpolarized to $\sim 7\%$ and consequently imaged in the reactor in situ or after liquid shuttling to a phantom. With a duty cycle of only 1 min and a lifetime of polarization of 1 min, the system provides a semiconstant supply of ^{13}C hyperpolarization. Using magnetic field cycling, $1\text{-}^{13}\text{C}\text{-EP}$ was hyperpolarized to almost 12.3% ^{13}C polarization. At high magnetic fields, the same ESOTHERIC approach as we used here achieved almost 60%;²¹ indicating that the concept for RF SOT polarization is valid. However, further hydrogenation optimization can improve the results as already discussed above.

12. In the section "Effects of low magnetic field on spin order transfer", the first paragraph is about field homogeneity (not field strength), which seems better suited for "Implications at inhomogeneous fields". I recommend moving this content to improve readability.

Answer:

In the modified paragraph we introduce the need for refocusing pulses and introduce parameters of refocusing.

Modified paragraph:

Effects of low magnetic field on spin order transfer. When the magnetic field B_0 is inhomogeneous, it is essential to refocus chemical shift evolution as rapidly as possible to prevent diffusion (or convection) induced dephasing of transverse magnetization in a nonuniform magnetic field⁶⁵⁻⁶⁷. Typical refocusing consists of two τ -intervals and 180° refocusing pulse in between (Figure 9b-d). This block can be repeated n times. The magnetic field can be inhomogeneous either on purpose (e.g., to prevent radiation damping^{49,68}) or due to imperfection of the hardware; the latter is the case here.

13. Please provide a comparison between the 45-CPMG method and the 45-OPE to verify that your 45-CPMG method is performing equivalently. This method is not validated for the reader.

Answer:

We overlaid data from Figure 3 on Figure 2. In principle the performance is identical because we changed the position of acquisition window only.

Figure 2 was changed and captions for Figures 2 and 3 were changed.

14. Field inhomogeneity – provide reference values of field homogeneity for your 0.55T system, compared to other polarizers, to help contextualize.

Answer:

We are not sure that it is of much interest since as the very high magnetic fields are sometimes suffering from high resolution and deliberately destroy magnetic field homogeneity see text:

Results:

When the magnetic field B_0 is inhomogeneous, it is essential to refocus chemical shift evolution as rapidly as possible to prevent diffusion (or convection) induced dephasing of transverse magnetization in a nonuniform magnetic field^{65–67}. Typical refocusing consists of two τ -intervals and 180° refocusing pulse in between (Figure 9b-d). This block can be repeated n times. The magnetic field can be inhomogeneous either on purpose (e.g., to prevent radiation damping^{49,68}) or due to imperfection of the hardware; the latter is the case here.

We have sufficient homogeneity for signal acquisition and quantification as demonstrated with e.g., Figure 1. At ultra-low fields, the homogeneity can be better than at superconducting magnets, but it does not make one of them a better polarizer. We think that homogeneity is not a proper characteristic for polarizers but for spectrometers. We do not see any benefits of using weak, selective pulses since it is not clear if they will give any better results.

E.g. in results:

To date, dozens of SOT sequences were proposed^{58,61–63}. Commonly, the SOT sequences with adiabatic or selective pulses give higher polarization for the selected nucleus. In our case, however, the magnetic field drifted about 100 Hz, or 5 ppm for ¹H (SI, Figure S1) in an hour, making such sequences impractical.

Typos

15. Typo: SOT defined twice

Answer:

Corrected.

Reviewer #3 (Remarks to the Author):

Ellermann et al. present a compact automated polarizer to prepare hyperpolarized molecules using parahydrogen-induced polarization (PHIP) based on a low-cost MRI scanner. They convert the anti-phase signal of the singlet ^1H spin pair into an in-phase signal using an echo with refocusing composite pulses to acquire the hyperpolarized signal with a better sensitivity despite the low homogeneity of the magnet. They report ^1H polarizations on the order of 15%. Their automated setup allows them to record a large data set of signal intensity vs echo time and number of echoes, which compares well with spin dynamics simulation. They show that the contribution of field inhomogeneity to losses during the echo is not significant for the echo times that they typically use. Finally, they use their setup to study the kinetics of hydrogenation for acetate and pyruvate precursors.

The results are interesting and will certainly benefit the parahydrogen community. Indeed, as pointed out by the Authors, automation is an important step towards more repeatable and reproducible PHIP experiments.

Answer: First, thanks for your genuine efforts to make the manuscript and the message more interesting for the reader.

A strong weakness of the paper is that the presented instrument and its performance are not compared with previously reported examples. If other polarizers are indeed mentioned in the introduction, a paragraph in the discussion or conclusion should be devoted to this comparison. What is the difference between the polarizer presented in Ellermann's paper compared to previously reported ones? How does the performance compare with those (eg: optimal ^1H polarization, duty cycle, cost, footprint,...)? Are the measured rates in Figure 5 more accurate than when measured with the good old shake-and-run experiment? What are the perspectives for improving the hardware presented here? Do the Authors intend to implement strategies to transfer the polarization to ^{13}C ? This is after all the strongest driving force for PHIP in MRI. Finally, I am no expert on the topic but I suspect that other automated systems exist that are not cited (eg. <https://doi.org/10.1002/cphc.202100667>)

Answer:

To address your point, we reworked the manuscript extensively. Moreover, we performed additional experiments. In detail, we did the following:

- We added one more section and one more figure with ^{13}C polarization transfer and imaging.
- We changed the introduction to give more context. (this reference <https://doi.org/10.1002/cphc.202100667> (now ref 36) and more others were added)
- We revised the conclusion.

New text in introduction:

In the literature, several approaches for PHIP are described, from "shake and run" with an NMR tube³¹ to complex, stand-alone^{17,32} or in situ polarizers for MRI³³ or NMR^{22,26,34-36}.

The very first PHIP polarizers for MRI contrast agents relied on spin order transfer (SOT) from parahydrogen via strong spin-spin interactions between ^1H and ^{13}C during MFC at fields below

0.1 mT^{37,38}. Soon after, polarizers with RF induced SOT were introduced³⁹ which doubled the experimentally achieved polarization reaching ~50% ¹³C polarization for partially deuterated 2-hydroxyethylpropionate, while only 21-25% was possible with MFC⁴⁰. Hyperpolarized 2-hydroxyethylpropionate is a non-active metabolic tracer and therefore has been used for MRI angiography instead³⁷. There are some more low-field systems with RF induced SOT which have been described⁴¹⁻⁴⁴.

In general, ultra-low (< 1 mT) and low-field (1 mT – 1 T) PHIP polarizers have the advantage of small physical size, low setup and running costs, and the fact that there is no need for cryogenics^{45,46}. So far, some of the highest polarization values with the hydrogenative PHIP-SAH approach, have been demonstrated using RF induced SOTs with superconducting high-field (~7 T) magnets^{21,47}.

New text in discussion, p. 6:

In addition, the hydrogenation reaction is possibly still limited by the solubility rate of hydrogen in the solution to be hyperpolarized. The straightforward way to change it would be using thicker tubings or stronger flow. In the literature, however, alternative methods were reported that should also be considered such as: semipermeable membrane reactors ideal for low B_0 magnetic field disturbance and continuous polarization^{36,73,74}, as well as ultra-sonic spraying of the solution to be polarized into a reaction chamber pressurised with pH_2 ⁷⁵.

New text in results and discussion, p. 6:

¹³C polarization and imaging. Using the same experimental conditions and ESOTHERIC SOT (Figure 9d) we hyperpolarized 1-¹³C-EP-d6 (Figure 6). Using numerical simulations, we found optimal parameters for the time intervals in the SOT for different number of refocusing pulses (SI, Figure S17). An experimental maximum of ~7% ¹³C polarization was achieved for two chemical shift refocusing pulse-circles. The hyperpolarization lifetime was about 1 min. Note that with a production duty cycle of about 1 min, it has the potential for continuous production of ¹³C hyperpolarization. This polarization of 50 mM ¹³C labelled tracer is equivalent to ~ 6000 M of ¹³C urea (Figure 6a); ¹H concentration in water is only ~110 M.

This level of polarization is already sufficient for multiple consecutive image acquisitions (Figure 6b,c). Using FLASH⁷⁶ MRI sequences, we acquired an image of the reaction chamber in situ (Figure 6b) and the image of hyperpolarized 1-¹³C-EP-d6 injected into a phantom (Figure 6c,d). As a phantom, we used a 3D printed negative of the "rotes Ampelmännchen" (red light pedestrian traffic symbol popular in Berlin, Germany) placed into a straight flat-bottom 10 mm NMR tube (Figure 6d).

A FLASH sequence with a 5° excitation angle, cartesian encoding without acceleration, an image matrix of 32 x 32, a field of view of 10 mm x 10 mm, a 10 mm slice thickness, and a repetition time of 50 ms allowed for 10-12 ¹³C images in situ of the reactor or the phantom until the signal could not surpass the sensitivity level anymore.

Text in conclusion:

The use of higher temperatures, a higher H₂ flow or more advanced H₂ mixing can improve it^{33,36,75}. The great advantage of a half-tesla benchtop is that it is (i) completely independent from the MRI or NMR system, (ii) portable with a footprint of 1 m², and (iii) after moving the setup to a new place it needs no more than a couple of minutes for automatic magnetic field calibration before usage. The manufacturer (Pure Devices GmbH) can provide magnets with a bore's inner diameter of 20 mm that will increase the hydrogenation volume from 300 μL to more than 3 mL. This increase in volume will be already comparable with preclinical dDNP systems^{77,78}.

If the manuscript is overall well structured, it is not easy to read. First, the text is split into too many paragraphs, sometimes of a single sentence. This is disruptive to the flow and makes it hard to follow the ideas that the Authors try to convey. Second, several variables and concepts are only defined in the methods section. The reader can therefore not always understand the main message (not only the details) without jumping to the end of the paper. Finally, there are many small mistakes in the language and the figure captions. I've tried to list as many as possible in the detailed comments below.

I suggest publishing this manuscript only after these two points above (comparison with the literature and readability) have been improved.

Answer:

Thank you for your critical points. As you see, we significantly changed the text to provide more context and improve the readability.

Throughout the manuscript

- "Polarization per proton" is a surprising concept; at least, it is unclear to me. Polarization does not need to be defined as "per proton". It is in itself relative. Consider defining it more precisely or calling it simply "polarization".

Old text, Figure 2, caption:

The estimated net-signal enhancement was 14.4% per proton (SI, Figure S4), with a theoretical maximum of 50%.

New text:

The estimated polarization of each proton was 14.4% (SI, Figure S4), with a theoretical maximum of 50% polarization for a two spin system⁶⁴.

And the same was corrected in SI in multiple places, e.g.:

Old text:

[...] polarization per proton 14.4% and enhancement 75250.

New text:

[...] average polarization of protons was 14.4% that corresponds to average enhancement of 75250.

- The Authors sometimes use the word "polarizer" and sometimes "hyperpolarizer". The former is more usual. Whatever they choose, the Authors should stick to one word
Abstract

Answer:

We consequently stick to the term "polarizer" now.

- "we demonstrated a repetitive hyperpolarization".
Replace "repetitive" with "repeatable"?

Old text:

With a setup footprint of 1 m², we demonstrated a repetitive hyperpolarization of ethyl acetate-d₆ and ethyl pyruvate-d₆ of 14.4 and 16.2% ¹H polarization correspondingly with a fully automated duty cycle of 1 minute.

New text:

With a setup footprint of 1 m², we demonstrated a semi-continuous fully automated hyperpolarization of ethyl acetate-d₆, ethyl pyruvate-d₆ of 14.4% and 16.2% ¹H polarization, respectively, and 1-¹³C-ethyl pyruvate-d₆ of 7% ¹³C polarization. The duty cycle for preparation of hyperpolarization is as short as 1 min.

Introduction

- “The very first PHIP-polarizers for biomedical ¹³C-MRI operated in this regime, and some more were described ever since^{41–45}.”

Add a reference after “in this regime”. Even if the reference is probably in 41-45. Adding the specific reference here would improve the clarity

Answer:

Thank you. We decided to give more context and changed complete section a bit.

New text:

In the literature, several approaches for PHIP are described, from "shake and run" with an NMR tube³¹ to complex, stand-alone^{17,32} or in situ polarizers for MRI³³ or NMR^{22,26,34–36}.

The very first PHIP polarizers for MRI contrast agents relied on spin order transfer (SOT) from parahydrogen via strong spin-spin interactions between ¹H and ¹³C during MFC at fields below 0.1 mT^{37,38}. Soon after, polarizers with RF induced SOT were introduced³⁹ which doubled the experimentally achieved polarization reaching ~50% ¹³C polarization for partially deuterated 2-hydroxyethylpropionate, while only 21-25% was possible with MFC⁴⁰. Hyperpolarized 2-hydroxyethylpropionate is a non-active metabolic tracer and therefore has been used for MRI angiography instead³⁷. There are some more low-field systems with RF induced SOT which have been described^{41–44}.

In general, ultra-low (< 1 mT) and low-field (1 mT – 1 T) PHIP polarizers have the advantage of small physical size, low setup and running costs, and the fact that there is no need for cryogen^{45,46}. So far, some of the highest polarization values with the hydrogenative PHIP-SAH approach, have been demonstrated using RF induced SOTs with superconducting high-field (~7 T) magnets^{21,47}.

The efficiency of SOT is limited if there are strong indirect interactions between different nuclei (¹H, ²H, ¹³C, etc.)^{48,49} and often 100% polarization is not accessible even theoretically with MFC^{40,50,51}. Ultra-low field conditions with MFC can be used for polarization of some molecules, however, in general they do not guarantee high polarization values and RF induced SOT is more flexible in this regard^{52–54}. Therefore, we opted for the latter approach.

- “The system uses a commercially available, portable 0.55 T "teaching" MRI based on permanent magnets.”

An “MRI” is not an instrument. The Authors should probably say “MRI scanner” or “MRI instrument”

New text:

The system uses a commercially available, portable, half-tesla benchtop "teaching" MRI scanner based on permanent magnets (detailed in methods, **Figure 8**).

- "To make use of ^1H polarization generated by $p\text{H}_2$, we used an out-of-phase spin echo (OPE) sequence^{47,48}, [...]"

The Authors should refer the reader to the methods section and figure 8 in particular?

New text:

To make use of ^1H polarization generated by $p\text{H}_2$, we implemented an out-of-phase spin echo (OPE) sequence (see methods, **Figure 9**)^{57,58}, which has recently been used for gas imaging⁵⁹. This sequence is of particular interest at inhomogeneous magnetic fields, since it converts hyperpolarized anti-phase spectra to in-phase NMR spectra, yielding a substantial increase in sensitivity of hyperpolarized NMR.

- A detailed summary of the content of the paper would be welcome in the end of the introduction.

New text:

Our modifications of the OPE sequence enabled real-time spying on the spin evolution without depletion of hyperpolarization. With this technique, we reduced the consumption of the sample to be polarized by 100 times, since complete spin evolution of anti-phase to in-phase conversion was measured using one sample only. With this approach, we studied hydrogenation reactivity of vinyl acetate and vinyl pyruvate: two popular PHIP-SAH precursors. This is one of the multiple steps that has to be optimized on the path towards in vivo applications. As a demonstration of the potential of our automated PHIP polarizer, we hyperpolarized 1- ^{13}C -ethyl pyruvate- d_6 to $\sim 7\%$ ^{13}C polarization using ESOTHERIC⁴⁷ (efficient spin order transfer to heteronuclei via relayed inept chains) SOT and acquired ^{13}C images of the reactor and a phantom. Ethyl pyruvate itself can be used as a metabolic MRI tracer either directly⁶⁰ or, after cleavage of the side arm, as pyruvate²¹. With further optimization and purification from the organic solvent and catalyst²¹, automated and rapid in vivo applications are granted.

Results

- "Some applications and findings on spin evolution and chemical reactions enabled by this setup are discussed below."

Is this sentence useful at all?

Answer: Agreed. Sentence was removed.

- "Because this spectrum is anti-phase, partial line cancellation occurs when B_0 is inhomogeneous."

Replace "spectrum" by "doublet"?

New text:

A pair of anti-phase inhomogeneously broadened PASADENA doublets was observed with a 45° -FID sequence (**Figure 2a-2**, methods: **Figure 9a**).

- "At high magnetic fields, [...] and the optimal τ -interval is given by $\tau=1/4Jn53$ "
Tau and n were not defined. Yes, they are defined in Fig. 8 but it would be more convenient for the reader to have the definition as they first appear in the text.

New text:

When the magnetic field B_0 is inhomogeneous, it is essential to refocus chemical shift evolution as rapidly as possible to prevent dephasing of transversal magnetization^{65–67}. Typical refocusing consists of two τ -intervals and 180° refocusing pulse in between (Figure 9b-d). This block can be repeated n times. The magnetic field can be inhomogeneous either on purpose (e.g., to prevent radiation damping^{49,68}) or due to imperfection of the hardware; the latter is the case here.

- "Although, in this case, we lose spectral resolution, we can follow (or "spy" on) the nuclear spin evolution over many echo intervals."
How did the Authors compute the signal intensity? Did they Fourier transform the signal or used the maximum of the FID?

New text:

We set the middle of the acquisition window in the middle of the expected echo and measured 9 data points at a 25 kHz rate. Although the amplitudes were very similar, the absolute maximum point was attributed to the signal of the echo. By measuring the amplitude of the echo, we obviously do not have spectral resolution, however, we can instead follow (or "spy" on) the nuclear spin evolution over many echo intervals.

- "Using this approach, we needed only one sample to measure one line of a 2D map (Figure 3c,d, and SI, Figures S2, S3)."
When the Authors mention "only one sample", do they mean "a single 300 μ L injection"? It is not completely obvious unless one reads the methods section in full.

New text:

By measuring the amplitude of the echo, we obviously do not have spectral resolution, however, we can instead follow (or "spy" on) the nuclear spin evolution over many echo intervals. Using this approach, we needed only one 300 μ L injection of the sample to measure one line of a 2D map (Figure 3c,d, and SI, Figures S2, S3). Using the 45° -OPE approach, we would need one sample injection per data point. As a consequence, we would need 30 mL of the sample to follow the spin evolution over the course of 100 spin-echoes. Instead, this was measured using a single sample of 0.3 mL with 45° -CPMG. Deuterated solvents (as done here) or additional signal-suppressing pulse sequence elements^{70,71} result in the echo signal being dominated by the signal from the hyperpolarized spins. The experimental observations are a good representation of the simulations.

- "Deuterated solvents (as done here) or additional signal-suppressing SOT elements result in the echo signal being dominated by the signal from hyperpolarized spins."
This statement deserves either a reference to published work or to a figure/section of the present manuscript.

Answer:

We added PASADENA specific filtering SOTs and general thermal signal suppression techniques.

- “We did not find means to improve the stability of the system.”

There are obvious means to improve the stability (eg. better insulation, PID-controlled heating) but they might be very difficult to implement. Consider rephrasing. Eg: “We did not try to further improve the stability of the system.”

Answer:

Indeed. PID was implemented by the manufacturer. Better insulation is possible but indeed takes a lot of time and either increases the size of the system substantially or forces us to open the system, which we prefer not to do. However, we are sure that we will address this issue one way or the other.

New text:

We did not **try to further improve** the stability of the system.

- “For echo intervals longer than $\tau \leq 170$ ms, the observed R_2^{obs} deviates from $1/T_2 + D^* (2\tau)^2$ dependence.”

Suggesting explanations for the observed deviation from the proposed model would be welcome.

Answer:

A plausible explanation is that Eq. 1 is valid for inhomogeneous but static magnetic fields which is not our case. On the longer τ -intervals the magnetic drift starts playing a more significant role. This can be one of the reasons of this deviation.

- “Note that for all τ in Figure 3, the observed relaxation times did not deviate much from the intrinsic T_2 value (Figure 4) [...]”

Consider specifying “for $\tau < 20$ ms”

New text:

Note that for all τ , which **were below 20 ms in the 45°-CPMG experiment (Figure 3)**, the observed relaxation times did not deviate much from the intrinsic T_2 (Figure 4).

- “High-resolution NMR allows for the observation of individual molecules and their transformations.”

“Chemical species” would be more appropriate than “molecules”. NMR only detects ensembles of molecules.

Answer:

We **changed** it **as** suggested.

- Eq. 2: Isn't H₂ missing in the reagents?

Answer:

Hydrogenation is a trimolecular reaction with a constant amount of catalyst Rh and dynamically constant amount of H₂ since we continuously bubble H₂. Because they are constant, the chemical

equation effectively has a pseudo-first order with a reaction rate of $k_1=[H_2][Rh]k_3$. This is indicated in eq 2 and in the text below:

See text:

where k_3 is the trimolecular reaction rate, $k_1 = [H_2][Rh]k_3$ is a pseudo-first-order rate because one can assume that the concentrations of the catalyst [Rh] and hydrogen [H₂] are constant during the entire hydrogenation experiment.

We hope that this is now clear.

- Eq 4: As far as I understand, here tau represents time and not the echo time. Consider replacing it with t to avoid confusion

Answer:

To make it in accordance with Figure 5 it is now τ_b

New text:

$$[V_{HH}^*] = \frac{[V_0]P_0}{1-R/k_1} (e^{-\tau_b R} - e^{-\tau_b k_1}) \quad (\text{Eq. 4})$$

where τ_b is the H₂ bubbling time and [V₀] is the initial concentration of the precursor V.

Figures

- Figure 2: 14.4% is the polarization and not the enhancement.

New text:

The estimated polarization of each proton was 14.4% (SI, Figure S4), with a theoretical maximum of 50% polarization for two spin system⁶¹.

- Figure 2: Where and how was the theoretical maximum computed?

Answer:

Reference was added. This is correct for two spin system.

New text: The estimated polarization of each proton was 14.4% (SI, Figure S4), with a theoretical maximum of 50% polarization for a two spin system⁶⁴.

- Figure 2b: It would help the readability to have first a precise list of what the figure contains (red circles and black dotted line) before it is interpreted.

- Figure 2b: I gather that the “highlighted” point is a red hollow circle replaced by a star. However, it took several minutes before realizing that this symbol was a highlighted point and not a cluster of red circles. Consider replacing it with a more obvious symbol.

Answer:

We added two arrows which indicate the spectrum and the polarization data point. Moreover, we modified the caption.

New caption text:

FIGURE 2. Ethyl pyruvate hyperpolarized at 0.55 T. Conversion of anti-phase EP-*d6* spectrum (a-2) into an in-phase spectrum (a-3) resulted in a boost of SNR: SNR was about six times higher in (a-3) than in (a-2). (b) 45°-OPE polarization transfer kinetic (red cycles – experimental, and black dashed line - simulations) helped us to find optimum parameters for the 45°-OPE(3); red arrows highlight the polarization value and corresponding spectrum (a-3). Blue small diamonds (b) are signal intensities acquired with 45°-CPMG SOT introduced further in the text and detailed in **Figure 3**. The in-phase spectrum is easier to analyze and has less impact from the low magnetic field homogeneity. The estimated polarization of each proton was 14.4% (SI, **Figure S4**), with a theoretical maximum of 50% polarization for a two spin system⁶⁴. Note that a new 300 μL sample injection was required for each data point on (b).

- Figure 2b: “[...] for phenomenological consideration of signal decay (black dots).”
I don't see black dots in the figure. Do the Authors mean the black dotted line?

New text:

The simulations for the observed net polarization of EP-*d6* (b) were multiplied by $\exp(-\tau/60\text{ms})$ for phenomenological consideration of signal decay (black dashed line).

- Figure 3 a and b: Consider specifying that the polarization is expressed between -1 and 1 using (-) to avoid confusion with %

Answer:

We changed the scale to % values and changed “polarization per proton” to “1H net polarization”.

Figure 3 was updated.

- Figure 4: As far as I understand, the red line corresponds to $R_2^{\text{obs}} = 1/T_2 + D^* (2\tau)^2$ and not to a monoexponential fit

Answer:

Correct.

New text:

Using 90°-CPMG (**Figure 9c**) 169 echoes were measured for each τ and signal decay was fitted with a monoexponential decay function to get R_2^{obs} (Eq. 1). Observed transversal relaxation rates were fitted with $R_2^{\text{obs}} = \frac{1}{T_2} + D^*(2\tau)^2$ for $\tau \leq 170$ ms (red line) resulting in $\frac{1}{T_2} = 0.262 \pm 0.001$ s⁻¹ or $T_2 = 3.81 \pm 0.01$ s and $D^* = 13.4 \pm 0.2$ s⁻¹. Sample: the same 10 mm tube with 300 μL of acetone-*d6*; the entire sample was inside the B_1 coil.

REVIEWER COMMENTS

Reviewer #2 (Remarks to the Author):

Thank you for your careful response to my previous suggestions. My only remaining comment is that, following revision, the Introduction is quite long. The Introduction should focus on the context needed to understand the novelty and the results, and some of the additional details would be better suited as Discussion points.

Reviewer #3 (Remarks to the Author):

I appreciate the efforts the Authors made to improve the readability of the paper and to address the issues I raised in the first round of revision. All the specific issues were addressed by the Authors' revisions. There is just one important point that could still be improved about the comparison of their polarizers with reported ones. The Authors have indeed added references and brief descriptions of reported polarizers but one sentence stating precisely what their instrument brings in comparison is missing. They state the performance of their polarizer. But how does it compare to existing ones? It does not need to be long but I still believe that this would increase the impact of the paper. If the Authors have reached the limit of words, I suggest moving some of the technical information to the supplement.

In addition to this point, I have noticed a few more elements that are worth improving (some that were already present in the previous version but that I noticed only now).

- "There are some more low-field systems with RF induced SOT which have been described." More details about these other polarizers would be welcome.

- "This condition depends on the spin system and nuclei under consideration⁴⁴ but is often met for fields above 1 mT." Do the Authors mean "prediction" instead of "condition"?

- " "Spying" on nuclear spins. Moreover, the polarization transfer deviates from the sine shape" Starting a section with "moreover" is unusual

- " As a result of hydrogenation, the product gains polarization P_{\square} . Under these assumptions, one can derive the evolution of VHH^* " Consider adding "as a function of the bubbling time" for more clarity

- "Hydrogenation kinetics for vinyl acetate to ethyl acetate (VA-d6 \rightarrow EA-d6) and vinyl pyruvate to ethyl pyruvate (VP-d6 \rightarrow EP-d6) (Figure 6) were measured and fitted using Eq. 4 (Figure 5)." The reference to Figure 6 seems incorrect

- "This polarization of 50 mM ¹³C labelled tracer is equivalent to \sim 6000 M of ¹³C urea (Figure 6a); ¹H concentration in water is only \sim 110 M." The term "polarization" seems inappropriate here. This is the magnetization of the hyperpolarized tracer at 50 mM compared to that of the 6000 M ¹³C urea at thermal equilibrium in the same conditions.

- " Using FLASH76 MRI sequences, we acquired an image of the reaction chamber in situ (Figure 6b) and the image of hyperpolarized 1-¹³C-EPd6 injected into a phantom (Figure 6c,d)." In the case of the liquid shuttling experiment, it is unclear how the experiment was performed since the polarizer and the MRI scanner are the same instrument. Was the sample hyperpolarized in the polarizer/MRI scanner and then the sample was replaced manually with the phantom filled by hyperpolarized solution?

REVIEWER COMMENTS

Reviewer #2 (Remarks to the Author):

Thank you for your careful response to my previous suggestions. My only remaining comment is that, following revision, the Introduction is quite long. The Introduction should focus on the context needed to understand the novelty and the results, and some of the additional details would be better suited as Discussion points.

Answer: Thank you for acknowledging that we have addressed all your comments! Thanks, too, for your new advice. We shortened the introductions by 200 words, added some discussion, and believe that the manuscript is much more concise now.

Reviewer #3 (Remarks to the Author):

I appreciate the efforts the Authors made to improve the readability of the paper and to address the issues I raised in the first round of revision. All the specific issues were addressed by the Authors' revisions.

Answer: many thanks!

There is just one important point that could still be improved about the comparison of their polarizers with reported ones. The Authors have indeed added references and brief descriptions of reported polarizers but one sentence stating precisely what their instrument brings in comparison is missing. They state the performance of their polarizer. But how does it compare to existing ones? It does not need to be long but I still believe that this would increase the impact of the paper. If the Authors have reached the limit of words, I suggest moving some of the technical information to the supplement.

Answer: Thanks for the advice, which we gladly incorporated into the manuscript. We tried to keep the overall length like before by streamlining the text and removing redundancy.

The following text was added on page 6:

New text: Most polarizers for hydrogenative ^{13}C PHIP were reported to operate at low field (2 – 50 mT)^{40,41,43,46,78}, high field (7 T – 9.4 T)^{21,33,53,79} or using field cycling (μT – T); and very few at intermediate regime used here, e.g., by Korchak *et al.*⁶⁹. The devices managed to attain a ^{13}C polarization of up to several tens of percent, although there was significant variation in yield, tracers, and concentration. Amongst the latest approaches for PHIP-SAH, Mammone *et al.* polarized 100 μL of 55 mM $^{13}\text{C}_2$ -cis-cinnamyl pyruvate ester-*d*2 in a 22.6 mT electromagnet to 24% of ^{13}C polarization⁴⁴, and Marshall *et al.*, polarized ^{13}C -cis-cinnamyl pyruvate ester-*d* at 50 μT magnetic shield to 9.8% of ^{13}C polarization⁸⁰. These polarizations are higher than those achieved here. We attribute this finding mostly to the inhomogeneous magnetic field of the MRI system (which was not designed for NMR). The advantages of our system include a high degree of automation, a short duty cycle, and the ability to run parameter variation experiments without human intervention. A comprehensive review on polarizers for hydrogeantive PHIP was published recently⁴⁵. It should be noted that very promising approaches to polarize biomolecules, including pyruvate with SABRE are emerging at this moment⁸¹.

In addition to this point, I have noticed a few more elements that are worth improving (some that were already present in the previous version but that I noticed only now).

- “There are some more low-field systems with RF induced SOT which have been described.”
More details about these other polarizers would be welcome.

Answer: We agree and added some more discussion and references on the polarizers. See the previous answer and new version of the introduction. In addition, we modified the sentence which now reads as

New text: The first PHIP polarizers for MRI contrast agents used strong spin-spin interactions between ^1H and ^{13}C during MFC at fields below 0.1 mT^{37,38} to transfer spin order from pH_2 to ^{13}C . Soon after, spin order transfer (SOT) pulse sequences and dedicated low-field polarizers were introduced³⁹, doubling the experimental ^{13}C polarization to ~50% for partially deuterated 2-hydroxyethylpropionate, whereas only 21-25% was reached with MFC⁴⁰. Hyperpolarized 2-hydroxyethylpropionate is not an endogenous, metabolically active molecule like pyruvate and was used for MRI angiography instead³⁷. There are some more low-field systems with RF field induced SOT⁴¹⁻⁴⁴, which have recently been comprehensively reviewed⁴⁵.

- “This condition depends on the spin system and nuclei under consideration⁴⁴ but is often met for fields above 1 mT.”

Do the Authors mean “prediction” instead of “condition”?

Answer:

Old text: In high-magnetic fields, 100% polarization for various molecules (especially for selectively deuterated ones) is commonly predicted, and more than 10% is regularly achieved experimentally^{48,56}. This condition depends on the spin system and nuclei under consideration⁴⁴ but is often met for fields above 1 mT.

New text: So far, some of the highest PHIP-SAH polarizations (especially for selectively deuterated molecules) have been demonstrated using RF-induced SOTs inside superconducting high-field magnets^{21,53,54}. Simulations also predict that the highest polarization can be achieved between 1 mT to 1 T (depending on the spin system). Fortunately, such fields can be realized easily, e.g., with portable permanent magnets or electromagnets with small footprints, low cost, and low maintenance (e.g., not requiring cryogenics)^{55,56}. The low-field magnets and reactor, scalable in size, can be adjusted to fit the desired volume of hyperpolarized media. In rodent studies, a typical injection dose is 150 μL , while in human studies, it can be as high as 30 mL. Achieving polarization of volumes used in human studies is challenging and has not yet been demonstrated with superconducting high-field polarizers.

One of the latest additions to the family of PHIP polarizers are reactors operated within high-field MRI systems, known as SAMBADENA⁵⁷. While the hardware requirements for this approach are low, challenges include limited space for the reactor (and thus limited tracer volume), accessibility, safety, and MRI-compatible materials.

- "Spying" on nuclear spins. Moreover, the polarization transfer deviates from the sine shape" Starting a section with "moreover" is unusual

Answer: Thank you for pointing this out. "Moreover" was removed.

Old text: Moreover, the polarization transfer deviates from the sine shape (**Figure 2b**) but also depends non-linearly on the number of refocusing pulses (**Figure 4a,b**) which is also the consequence of the intermediate coupling between protons⁷⁰.

New text: A consequence of the intermediate coupling between the protons⁷⁰ is the non-sinusoidal polarization transfer (**Figure 2b**), which also depends non-linearly on the number of refocusing pulses (**Figure 4a,b**).

- "As a result of hydrogenation, the product gains polarization P_{\square} . Under these assumptions, one can derive the evolution of VHH*" Consider adding "as a function of the bubbling time" for more clarity

Answer: Agreed.

New text: Under these assumptions, one can derive the evolution of VHH* as a function of bubbling time (for details see SI of Ref. ²⁷):

- "Hydrogenation kinetics for vinyl acetate to ethyl acetate (VA-d6 → EA-d6) and vinyl pyruvate to ethyl pyruvate (VP-d6 → EP-d6) (Figure 6) were measured and fitted using Eq. 4 (Figure 5)." The reference to Figure 6 seems incorrect

Answer: Thanks!

New text: Hydrogenation kinetics for vinyl acetate to ethyl acetate (VA-d6 → EA-d6, **Figure 5a**) and vinyl pyruvate to ethyl pyruvate (VP-d6 → EP-d6, **Figure 5b**) were measured and fitted using Eq. 4.

- "This polarization of 50 mM ¹³C labelled tracer is equivalent to ~ 6000 M of ¹³C urea (Figure 6a); ¹H concentration in water is only ~110 M." The term "polarization" seems inappropriate here. This is the magnetization of the hyperpolarized tracer at 50 mM compared to that of the 6000 M ¹³C urea at thermal equilibrium in the same conditions.

Answer: Agreed.

Old text: This polarization of 50 mM ¹³C labelled tracer is equivalent to ~ 6000 M of ¹³C urea (**Figure 6a**); ¹H concentration in water is only ~110 M.

New text: Note that the magnetization (or signal intensity) of 50 mM ¹³C tracer hyperpolarized to 7% is approximately equal to ~6000 M thermally polarized tracer at 0.55 T (**Figure 6a**). This exceeds the concentration of water (~55 M) more than 100-fold.

- " Using FLASH76 MRI sequences, we acquired an image of the reaction chamber in situ (Figure 6b) and the image of hyperpolarized 1-¹³C-EPd6 injected into a phantom (Figure 6c,d)."

In the case of the liquid shuttling experiment, it is unclear how the experiment was performed since the polarizer and the MRI scanner are the same instrument. Was the sample hyperpolarized in the polarizer/MRI scanner and then the sample was replaced manually with the phantom filled by hyperpolarized solution?

Answer: Thanks for pointing that out. We clarified the respective paragraph.

New text: For the latter, we hyperpolarized 1-¹³C-EP-*d6* as before. Then, we transferred the liquid from the reaction chamber into a second receiver tube using the remaining pressure (via three-way valve M1, **Figure 8**). Once the liquid was settled in the receiver tube, we exchanged the reaction chamber and receiver tube and initiated image acquisition. The receiver tube, a 10 mm flat bottom NMR tube, contained a 3D printed negative of the "rotes Ampelmännchen" (red light pedestrian traffic symbol popular in Berlin, Germany, **Figure 6d**).

In both cases, high-resolution, fully-sampled cartesian ¹³C-FLASH MRI was acquired within 1.5 s, without any dedicated acceleration techniques (voxel size (312 μm)², 5° excitations, repetition time 50 ms, 10 mm slice). Hyperpolarized ¹³C signal was apparent on ten images acquired every ~ 5 s. Limitations of the gradient power prohibited us from using smaller voxels or larger matrices.

REVIEWERS' COMMENTS

Reviewer #3 (Remarks to the Author):

All the points I have raised at the previous stage have been appropriately addressed by the Authors.

I have no further comments and suggest publishing the manuscript as is.